# Crossmodal sensory neurons based on high-performance flexible memristors for human-machine in-sensor computing system

Zhiyuan Li[1,2,7], Zhongshao Li[3,4,7], Wei Tang[1], Jiaping Yao[1], Zhipeng Dou[5], Junjie Gong[1], Yongfei Li[1], Beining Zhang[1], Yunxiao Dong[1], Jian Xia[1], Lin Sun[5], Peng Jiang ®[5], Xun Cao ®[3,4] ✉, Rui Yang ®[1,2] ✉, Xiangshui Miao ®[1,2] ✉ & Ronggui Yang ®[6]

Constructing crossmodal in-sensor processing system based on high-performance flexible devices is of great significance for the development of wearable human-machine interfaces. A bio-inspired crossmodal in-sensor computing system can perform real-time energy-efficient processing of multimodal signals, alleviating data conversion and transmission between different modules in conventional chips. Here, we report a bio-inspired crossmodal spiking sensory neuron (CSSN) based on a flexible $VO_2$ memristor, and demonstrate a crossmodal in-sensor encoding and computing system for wearable human-machine interfaces. We demonstrate excellent performance in the $VO_2$ memristor including endurance ($>10^{12}$), uniformity (0.72% for cycle-to-cycle variations and 3.73% for device-to-device variations), speed (<30 ns), and flexibility (bendable to a curvature radius of 1 mm). A flexible hardware processing system is implemented based on the CSSN, which can directly perceive and encode pressure and temperature bimodal information into spikes, and then enables the real-time haptic-feedback for human-machine interaction. We successfully construct a crossmodal in-sensor spiking reservoir computing system via the CSSNs, which can achieve dynamic objects identification with a high accuracy of 98.1% and real-time signal feedback. This work provides a feasible approach for constructing flexible bio-inspired crossmodal in-sensor computing systems for wearable human-machine interfaces.

To interact with the real world effectively, advanced robotics should be equipped with a crossmodal sensing-computing system that can perceive multimodal surroundings, recognize objects, and provide real-time feedback[1–3]. For example, when working in extreme and hazardous environments (e.g., surgery, ruins rescue, and submarine

survey), single-modal sensing systems of robotics have difficulty in executing complex tasks, and is even prone to harm to devices and users[4,5]. With crossmodal sensing and processing capabilities, intelligent robots could access comprehensive object features (e.g., color, shape, texture, softness, temperature, odor, and motion) and provide

[1]School of Integrated Circuits, Huazhong University of Science and Technology, Wuhan, China. [2]Hubei Yangtze Memory Laboratories, Wuhan, China. [3]State Key Laboratory of High Performance Ceramics and Superfine Microstructure, Shanghai Institute of Ceramics, Chinese Academy of Sciences, Shanghai, China. [4]Center of Materials Science and Optoelectronics Engineering, University of Chinese Academy of Sciences, Beijing, China. [5]State Key Laboratory of Catalysis, CAS Center for Excellence in Nanoscience, Dalian Institute of Chemical Physics, Chinese Academy of Sciences, Dalian, China. [6]State Key Laboratory of Coal Combustion, School of Energy and Power Engineering, Huazhong University of Science and Technology, Wuhan, China. [7]These authors contributed equally: Zhiyuan Li, Zhongshao Li. ✉e-mail: cxun@mail.sic.ac.cn; yangrui@hust.edu.cn; miaoxs@hust.edu.cn

real-time information feedback[6–8], which is of value in autonomous driving, environment exploration, and human-machine interactions.

State-of-the-art complementary metal-oxide-semiconductor (CMOS) sensing-computing systems have been used to realize complicated and dexterous multisensory processing and motion control functions[3,9,10]. However, these systems are in massive system designs with separated sensing-computing architecture and sophisticated analog-digital conversion, which causes serious latency in processing and inevitably increases the power consumption. In contrast, the biological multisensory system provides a crossmodal in-sensor computing paradigm, with multimodal fusion and parallel processing capabilities, facilitating robust and energy-efficient object recognition[11,12]. As shown in Fig. 1a, when a person touches an object, analog perception signals (e.g., pressure, temperature, etc.) can be preprocessed and synchronously encoded into neuronal trains through sensory neurons[13]. The spike-encode information is then conveyed to the cerebral cortex, where the information transmitted is postprocessed to accurately recognize object properties and avoid damage caused by dangerous signals.

Bioinspired in-sensor computing systems have emerged as a promising candidate for the multimodal perception and processing of analog signals from the physical world, alleviating data transmission bottlenecks across sensor-processor interfaces[14]. One of the emerging devices in this field is memristor[15,16], which works more efficiently than CMOS devices such as the synapses[17–19] and neurons[20–22], thanks to its simple two-terminal structures and abundant ion dynamics, allowing for in-sensor computing. Remarkably, the neuromorphic system based on memristors has been successfully demonstrated to sense and process visual[23–25], tactile[26–29], and auditory information[30,31]. However, realizing a flexible in-sensor computing system with crossmodal spike encoding and real-time haptic feedback capabilities remains challenging. To implement such a flexible system, several fundamental challenges need to be overcome. First, flexible memristor non-idealities, such as low durability, noticeable cycle-to-cycle (C2C) and device-to-device (D2D) variability, make them challenging to build high-performance neuromorphic systems on large-area flexible substrates. Second, it is difficult to achieve crossmodal in-sensor spike encoding by simply combining sensors with conventional memristors, due to the need of additional conversion circuits for signal acquisition

at the multiple sensor nodes, resulting in more time- and energy-consuming. Third, it is required that the flexible in-sensor computing system accurately identifies multimodal objects and provides real-time haptic feedback for real applications in human-machine interaction.

In this work, a flexible memristor-based crossmodal spiking sensory neuron (CSSN) is developed to process multimodal signals and provide haptic-feedback for in-sensor computing system (Fig. 1b). The CSSNs perform in-sensor spike encoding to capture and extract critical features from the crossmodal signals, like the sensory neuron does in the biological system. Then, the encoded information is delivered and classified in the spiking reservoir network which behaves like cerebral cortex in biological system, as shown in Fig. 1. The CSSN unit consists of two key components: a flexible temperature-sensitive memristor and a flexible pressure sensor. A forming-free flexible $VO_2$ memristor is fabricated at low temperature (280 °C) by introducing a $Cr_2O_3$ buffer layer, exhibiting excellent yield (~97.3 %), ultrahigh endurance (>$10^{12}$ cycles), low C2C and D2D variation (0.72% and 3.73%, respectively), fast respond speed (<30 ns), and high flexibility (bendable to a radius of 1 mm). This device can not only achieve neuron firing by threshold switching (TS) behavior but also sense the temperature by the intrinsic temperature-sensing capability of the $VO_2$ material. By coupling with a flexible pressure sensor, the CSSN can synchronously encode bimodal signals (e.g., pressure and temperature) as spikes with different frequencies. Furthermore, the CSSN can perform real-time haptic feedback to external signals by utilizing integrated, flexible hardware. By leveraging CSSNs as the feature extraction and fusion layer of multimodal signals, a crossmodal in-sensor spiking reservoir computing system is demonstrated for dynamic object identification and real-time signal feedback. Compared to single-modal sensory recognition (83.6% and 79.1% for sole pressure and temperature, respectively), the multimodal system shows higher recognition accuracy (98.1%) and exhibits more realistic sensing feedback under bimodal sensing fusion.

## Results

### High-performance flexible $VO_2$ memristor

The key device of the spiking in-sensor computing system, the flexible $VO_2$ memristor (Fig. 2a), was fabricated by depositing monoclinic (M) $VO_2$ on the $SiO_2$/polyimide (PI) substrate, followed by electron-beam lithography, metallization, and lift-off to define the planar electrodes.

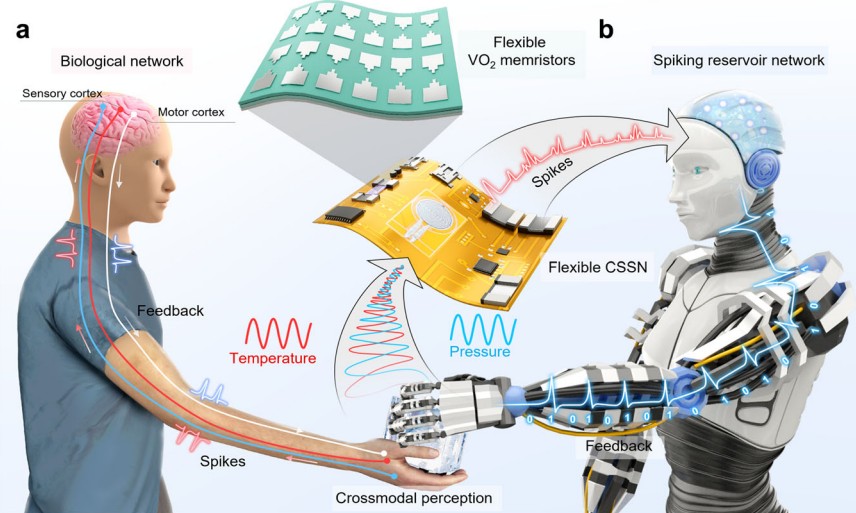

**Fig. 1 | Bio-inspired crossmodal intelligent in-sensor computing system.**
a Schematic diagram of the multisensory processing in biological proprioceptive system. **b** Schematic illustration of the spiking crossmodal in-sensor computing system. Integrated flexible crossmodal spiking sensory neurons (CSSNs) consisting of high-performance flexible $VO_2$ memristors and sensors. The CSSNs can

crossmodally encode signals (temperature and pressure) into spikes, working like the sensory neuron does in the biological system. Then, these spikes are input into a spiking reservoir network for crossmodal recognition and feedback. This spiking reservoir network behaves like the cerebral cortex in a biological system.

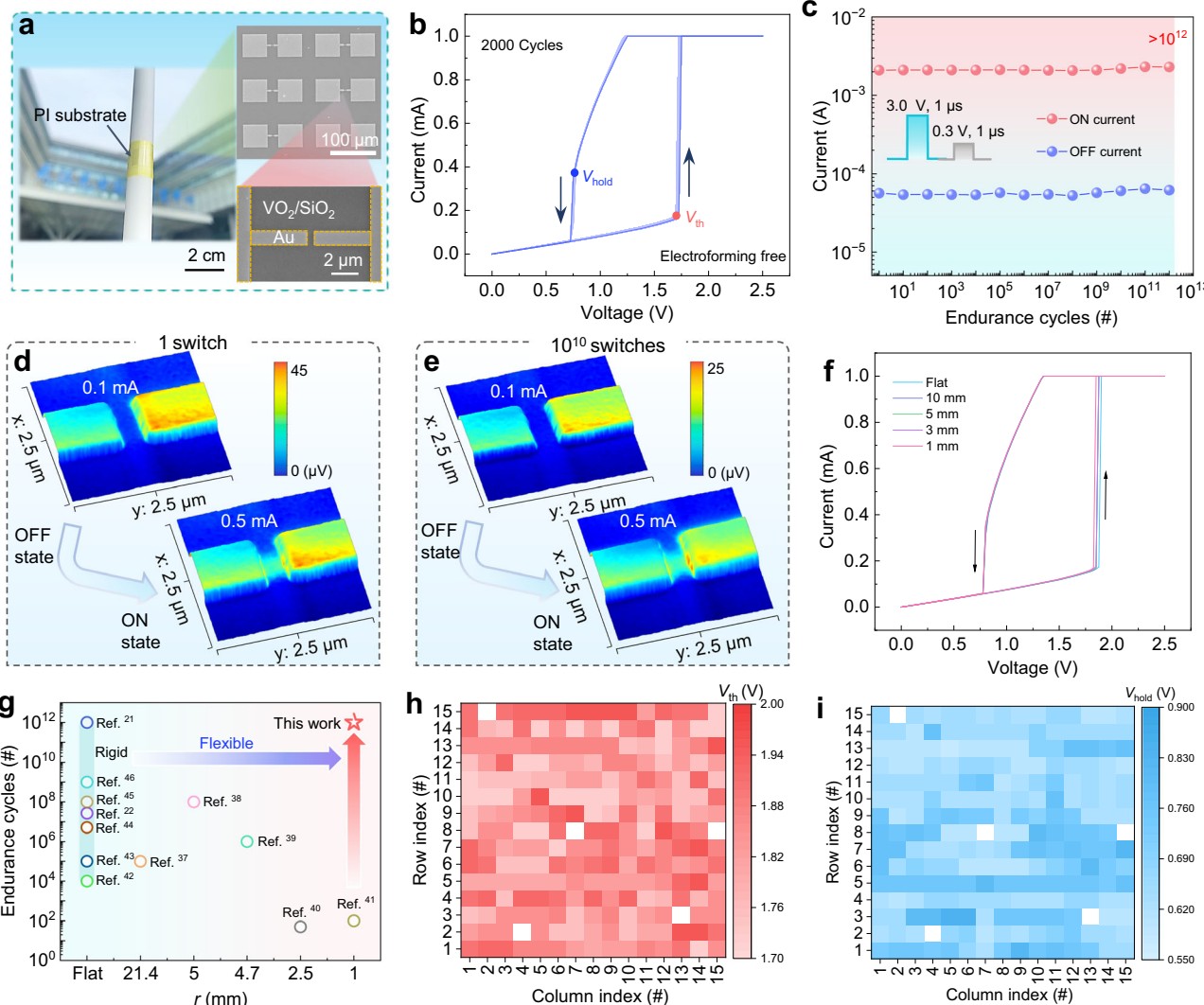

**Fig. 2 | Structure and electrical characterization of the flexible VO₂ memristor.** **a** Photograph of flexible VO₂ memristors developed in this work (top right: enlarged scanning electron microscope (SEM) image of the planar structure, bottom right: magnified SEM image of the channel with a length of 500 nm). **b** Typical threshold switching (TS) characteristic of the VO₂ memristor. The current-voltage (*I-V*) sweeps show no variation after the 2000 cycles. **c** The endurance test showing no degradation of signals over $10^{12}$ cycles by applying the pulse pair of 3.0 V/1 μs and 0.5 V/1 μs. In-situ scattering-type scanning near-field optical microscopy (s-

NOM) image of the device ON/OFF area under 1 switch (**d**) and $10^{10}$ switches (**e**). The morphology of the conductive channel did not show observable changes after $10^{10}$ switches. **f** The TS characteristics of the flexible VO₂ memristor under different bending radii (*r* from 10 to 1 mm). **g** Comparison of reported silicon-based[21,22,42–46] and flexible[37–41] neuronal switching devices with the present VO₂ memristor in terms of endurance and flexibility characteristics. The device-to-device variation of the threshold voltage ($V_{th}$) (**h**) and holding voltage ($V_{hold}$) (**i**) from 15 × 15 memristor matrix showing a 97.8% device yield.

The active area size of VO₂ memristor is a channel length of 500 nm and a width of 1 μm. More detailed fabrication process can be found in the Supplementary Fig. 1 and Methods. The metal-to-insulator transition (MIT) of VO₂ results in TS behavior of VO₂ memristors. To obtain excellent MIT features, VO₂ materials are typically grown at relatively high temperatures (> 450 °C)[23,32,33]. However, such high temperature is not suitable for the manufacturing of flexible electronics. To overcome this challenge, Cr₂O₃ was chosen as the buffer layer to lower the lattice mismatch between VO₂ and SiO₂, and to reduce the deposition temperature of VO₂ film to 280 °C. We optimized the buffer layer's thickness by preparing VO₂ thin films and devices with varying buffer layer thicknesses (0–40 nm), achieving an optimal thickness of 40 nm, as shown in Supplementary Figs. 2–7. It is found that the Cr₂O₃ buffer layer helps to improve the crystalline quality of the VO₂ (M) thin film, resulting in a stable MIT process, detail results are explained in Supplementary Note 2. The VO₂ memristor with a 40 nm-thick buffer layer not only exhibits forming-free behavior but also achieves high stability

and uniformity. These improvements promote practical applications and large-scale fabrication.

The switching characteristics of the VO₂ memristor were systematically studied. Figure 2b shows the quasi-static current-voltage (*I-V*) characteristics of the flexible VO₂ memristor. The device displays typical volatile TS behavior without an electroforming process. The forming-free behavior is primarily attributed to the high-quality VO₂ (M) thin films obtained by introducing a Cr₂O₃ buffer layer with optimal thickness, as shown in Supplementary Fig. 3. The operating principle of devices is the reversible MIT of the VO₂ gap. In the resting state, the memristor stays in an insulating phase (i.e., high resistance state (HRS)). When the applied voltage exceeds the threshold voltage ($V_{th}$), the output current increases abruptly due to the transition from insulator to metal, and then the device is in a low resistance state (LRS). Once the applied bias is smaller than the holding voltage ($V_{hold}$), the device spontaneously returns to HRS, as a result of the transition from metal to insulator. In-situ scattering-type scanning near-field optical

microscopy (s-SNOM) was used to observe the evolution of active conductive channel during the continuous transition between insulator and metal states (Supplementary Fig. 8). Notably, the changes in the SNOM signal observed in the channel area are primarily associated with the phase transition of the VO$_2$ thin film. This transition involves significant alterations in the local optical properties of the material[34]. After $10^{10}$ switches, there is no observable change in the conductive channel, see Fig. 2d, e. The stable conductive channel is mainly attributed to good crystalline quality and stable MIT of the VO$_2$ (M) film with Cr$_2$O$_3$ buffer layer. Furthermore, the switching endurance of the devices was examined over $10^{12}$ pulses. No degradation of ON/OFF state is observed during continuous operations (Fig. 2c), revealing excellent endurance of the device. Notably, an external compliance current of ~2 mA was set to safeguard the memristor from overcurrent damage under applying voltage pulse by incorporating a 1 kΩ resistor in series with the device. The switching endurance of VO$_2$ memristor is translatable to stable oscillating behavior in a neuronal circuit, which further ensures the reliability of artificial neurons that incorporate such devices, detail results are shown in Supplementary Fig. 9. The transient turn-on and turn-off response time of the device are measured to be about 30 and 50 ns, respectively (Supplementary Fig. 10). These excellent TS properties of the proposed memristor enable a reliable and fast neuronal spike-encoding in the CSSN.

For flexible devices, sufficient bending resistance, bending durability, high yield and low variations of devices are crucial prerequisites for the flexible electronic application[35,36]. Our flexible memristors were tested on a sliding fixture with different bending radii to study the mechanical flexibility (Supplementary Fig. 11a, b). The bending orientation is an out-of-plane bending, meaning the device is flexing along a curvature (Supplementary Fig. 11c). The device shows no change in TS performance (Fig. 2f) even when the bending radius ($r$) is decreased to 1.0 mm, which reaches the highest flexibility in the reported flexible TS devices[37–41] (Fig. 2g). During consecutive 1000 mechanical bending cycles ($r = 5$ mm), the electrical characteristics of the flexible VO$_2$ memristor show no deterioration (Supplementary Fig. 11d). The excellent robustness of flexible memristors under bending is primarily due to the negligible geometric change in the active area, facilitated by their small dimensions. Our stretching experiments revealed that these devices continue to function normally under tensile strains up to 10%, as shown in Supplementary Fig. 12. Achieving high yield and low variation of devices is much more challenging for flexible electronic devices compared to the rigid ones. We measured the electrical characteristics of 15 × 15 devices (Supplementary Fig. 13a, b). The present flexible devices exhibit a yield of 97.8 %, and an excellent uniformity for several important device metrics, including $V_{th}$ (Fig. 2h) and $V_{hold}$ (Fig. 2i). The D2D variation ($C_v = \sigma/\mu$, σ is the standard deviation and μ is the mean value) of the $V_{th}$ and $V_{hold}$ are 3.73% and 9.35%, respectively (Supplementary Fig. 13c). The comparisons of D2D variations with other reported TS devices are presented in Supplementary Table 2. The present device performs high yield and uniformity. The device also shows low C2C variation during 2000 consecutive $I$-$V$ sweeps with a compliance current of 1 mA, see Fig.2b. The C2C variation in $V_{th}$ and $V_{hold}$ is as low as 0.72% and 0.31%, respectively (Supplementary Fig. 14). The excellent D2D uniformity and low C2C variation mainly stem from the merits of electroforming free and good crystallization of the VO$_2$ (M) thin film with Cr$_2$O$_3$ buffer layer. Compared to reported neuronal switches, including state-of-the-art silicon-based devices[21,22,42–46] and flexible devices[37–41], the present flexible VO$_2$ memristor exhibits significant competitiveness and advantages in terms of endurance and flexibility, see Fig. 2g. The scalability of the flexible device is critical for a wide range of applications, particularly in the fields of scalable electronics and large-area sensing systems. To investigate the scalability of the flexible VO$_2$ memristor, $I$-$V$ characteristics of the devices with different sizes were

measured across a range of channel lengths (300 nm–4 μm) and widths (500 nm–4 μm), as shown in Supplementary Fig. 15. The results demonstrate that the $V_{th}$ decreases as the device size decreases, and there is no significant change in the TS stability of the device, indicating that the device is scalable without property degradation.

## Characterization of temperature and pressure sensing and encoding properties

Tactile perception is crucial in handling daily interactions with objects, safely manipulating and exploring objects to comprehensively understand the physical characteristics of an object. Biological sensory neurons within the flexible skin provide a simple and intuitive interface to encode the external temperature and mechanical stimuli into neuronal spike-trains for object recognition. Inspired by biological sensory neurons, we designed different artificial spiking sensing neuron circuits which can perceive and process temperature/pressure information into neuronal signals (Fig. 3a, d).

A leaky integrate-and-fire (LIF) neuron circuit based on a VO$_2$ memristor was adopted to realize the spike-encoding functionality of the spiking neuron (Supplementary Fig. 16 and Supplementary Note 2). To perceive and encode temperature information, the intrinsic temperature-dependent phase transition of the VO$_2$ memristor was used to design a temperature-sensing spiking neuron circuit based on the LIF circuit (Fig. 3a). As shown in Supplementary Fig. 17a, the $V_{th}$ of the device significantly decreases as the temperature increases. At the same time, the resistance of memristor ($R_{mem}$) slightly decreases. There is an inverse proportion relationship between the $V_{th}/V_{hold}$ and temperature, with the detailed results shown in Supplementary Fig. 17b, c. This temperature-dependent $V_{th}/V_{hold}$ directly affects the spiking behavior of the circuit. Therefore, this circuit based on VO$_2$ memristor can incorporate the thermoreceptor and spike-encoding properties to implement temperature-encoding and sending. In this circuit, the operation temperature ranges from 21 to 42 °C is chosen to achieve effectively sensing-encoding under different load resistor ($R_L$). Real-time spiking responses are shown in Fig. 3b and Supplementary Figs. 18, 19. The firing frequency ($f_{fri}$) characteristics of this circuit under different temperatures and load resistor ($R_L$) are shown in Fig. 3c. As shown in Fig. 3b, when the external temperature rises from 21 to 42 °C, spike trains show an increased firing frequency ($f_{fri}$) and a reduced voltage amplitude. According to the analysis of this circuit, the spike amplitude ($V_{out}$) is related to $V_{th}$, monitor resistor ($R_{out}$) and the LRS of the memristor ($R_{on}$) as follows (Eq. 1):

$$V_{out} = V_{th} \times \frac{R_{out}}{R_{out} + R_{on}} \tag{1}$$

Thus, the amplitude of $V_{out}$ mainly depends on $V_{th}$ as the temperature increases. To study the dependence of $f_{fri}$ on temperature, we first calculate the charging time ($t_{rise}$) and discharging time ($t_{fall}$) of the sensory neurons, which can be described by the following equations (Eqs. 2, 3):

$$t_{rise} = C_m \left( R_{rise} \| R_L \right) \times \ln \left( \frac{V_{in} \frac{R_{rise}}{R_L + R_{rise}} - V_{hold}}{V_{in} \frac{R_{rise}}{R_L + R_{rise}} - V_{th}} \right) \tag{2}$$

$$t_{fall} = C_m \left( R_{fall} \| R_L \right) \times \ln \left( \frac{V_{in} \frac{R_{fall}}{R_L + R_{fall}} - V_{th}}{V_{in} \frac{R_{fall}}{R_L + R_{fall}} - V_{hold}} \right) \tag{3}$$

where $R_{rise}$ and $R_{fall}$ are the $R_{mem} + R_{out}$ during the charging and discharging process, respectively. Furthermore, the $f_{fri}$ can be expressed

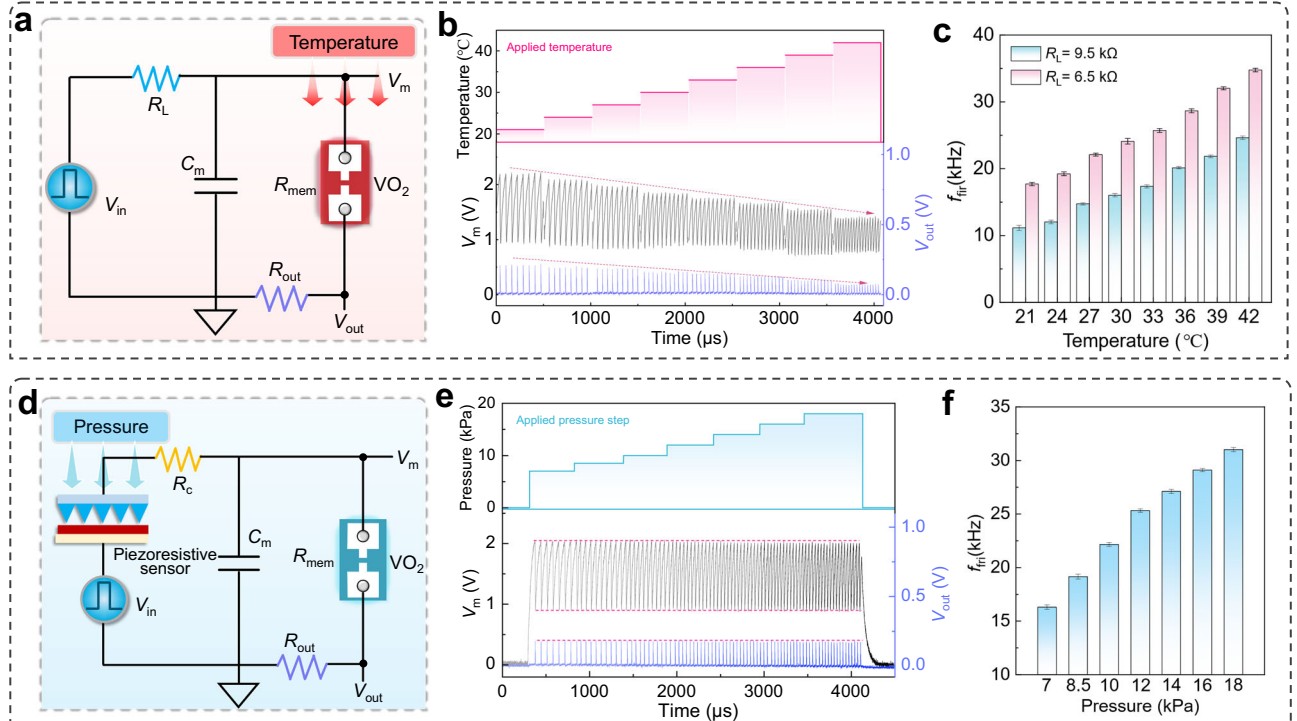

**Fig. 3 | Temperature and pressure sensing and encoding characteristics of the artificial spiking sensory neuron. a** Circuit diagram of the artificial spiking temperature-sensing neuron, $V_{in} = 5\,V/800\,\mu s$, $C_m = 10\,nF$, $R_L = 6.5/9.5\,k\Omega$, and $R_{out} = 50\,\Omega$. **b** Real-time spiking responses from the spiking temperature sensory neuron by applying different temperatures (21–42 °C). **c** The $f_{fri}$ of the spiking temperature sensory neuron under different $R_L$ and temperatures (21–42 °C). **d** Circuit diagram of the artificial spiking pressure-sensing neuron. $V_{in} = 5\,V/800\,\mu s$, $C_m = 10\,nF$, and $R_c/R_{out} = 3\,k\Omega/50\,\Omega$ were used in the circuit. **e** Real-time spiking responses from the spiking pressure sensory neuron by applying different pressures (0–18 kPa). **f** Output firing frequency ($f_{fri}$) of the spiking pressure sensory neuron under different pressures. $f_{fri}$ was obtained by averaging spike responses during six consecutive pressing-releasing cycles. The $f_{fri}$ was obtained by averaging five spike responses at the same temperature.

by Eq. 4:

$$f_{fri} = \frac{1}{t_{rise} + t_{fall}} \qquad (4)$$

After considering the influence of temperature, the threshold and holding voltage can be described as follows[47]:

$$V_{th}(T_0) = \sqrt{\frac{R_{off}}{R_{th}} \cdot (T_t - T_0)} \qquad (5)$$

$$V_{hold}(T_0) = \sqrt{\frac{R_{on}}{R_{th}} \cdot (T_t - T_0)} \qquad (6)$$

where $R_{th}$, $T_t$, and $T_0$ are the effective thermal resistance, the transition temperature of $VO_2$, and the operating temperature, respectively. Therefore, the impact of temperature on sensory neuron spiking can be obtained by inserting Eqs. (5) and (6) into Eqs. (1)-(4). For the temperature sensory neuron, the increase in $f_{fri}$ mainly stems from the decrease in $V_{th}$ and $R_{mem}$ caused by rising temperature[48]. Thus, temperature encoding characteristics can be obtained from the frequency and amplitude information of spiking responses. With the adjustment of the $R_L$, the temperature-sensing neurons can successfully realize the strength-modulated $f_{fri}$ characteristics, providing the basis for subsequent pressure-temperature crossmodal sensory design. Notably, our artificial temperature-sensing neuron shows a stable spiking response around 36 °C (Supplementary Fig. 19f), which is close to the surface temperature of the human skin, indicating

that the sensory neuron can potentially be used in real-time physiological temperature monitoring.

Based on the LIF circuit, an artificial spiking pressure-sensing neuron circuit was designed by integrating with a flexible piezoresistive sensor (Fig. 3d). In this circuit, the flexible piezoresistive sensor, functioning as a mechanoreceptor, detects pressure information; the $VO_2$ memristor directly converts the detected pressure information into output spikes. The piezoresistive performance of the pressure sensor is shown in Supplementary Fig. 20. The LIF neuron firing places a requirement on the range of the $R_L$. To meet neuronal circuit firing, a calibrated resistor ($R_c$) is connected in series to this circuit to adjust the pressure sensor that is too low in resistance ($R_p$) when applying high pressure. This $R_c$ also helps to attenuate the effect of device variation on the frequency of firing spike[23]. Based on this circuit, the artificial sensory neuron can directly respond to different pressure-stimuli and encode them into spike trains where real-time spiking responses are shown in Fig. 3e. As the external pressure increase, spike trains demonstrate an increased $f_{fri}$ and a constant voltage amplitude. Figure 3f displays the relationship between $f_{fir}$ and pressure. As applied pressure increase from 0 to 18 kPa, the $f_{fri}$ monotonically increases from 16.2 to 31.0 kHz, resulting from the overall increased charging speed with the resistance of $R_p$ decreasing. To further study the dependence of pressure on the $f_{fri}$, we extend the model to pressure-spiking sensory neurons, which can be described by the following equation:

$$R_L = R_c + R_p \qquad (7)$$

The impact of pressure on sensory neuron spiking can be obtained by inserting Eq. (7) into Eqs. (1)-(4). Notably, when neurons

encode both pressure and temperature information, the model needs to be built by combining Eqs. (5) and (6). For the pressure sensory neuron, the increase in $f_{fri}$ mainly stems from the decrease in $R_p$ caused by rising pressure. Thus, pressure encoding property can be obtained from the frequency information of spiking responses.

With increasing numbers of tactile data, there is an urgent need for the crossmodal encoding capability to efficiently process multi-modal data locally, which further alleviates the burden of sensors and sensor/processor interfaces[14,49]. As the above in-sensor temperature and pressure encoding capabilities of artificial spiking sensing neurons, it provides a potential solution for in-sensor crossmodal encoding. The circuit diagram of the CSSN is the same as the pressure-sensing neuron circuit, as shown in Fig. 3d. In this circuit, the temperature-sensitive $VO_2$ memristor offers a temperature sensing while the piezoresistive sensor provides pressure sensing, and temperature and pressure signals can be synchronously processed into the spike trains with different $f_{fri}$ and amplitude by crossmodal in-sensor encoding process of the CSSN. Therefore, the extra temperature sensors are not needed for this circuit configuration, significantly reducing the circuit complexity and improving the encoding efficiency. Notably, since temperature encoding can cause changes in spike amplitude, while pressure encoding does not, thus the bimodal information can be decoupled by the CSSN. These features enable CSSN with excellent data compression and conversion capabilities, eliminating severe signal interference and complex decoupling processes.

High energy efficiency is critical in artificial sensory systems. To validate the efficiency of CSSNs in constructing high-efficiency human-machine interfaces, we further investigated its energy consumption per spike. We achieved a minimum of ~3.9 nJ per event with optimal parasitic capacitance settings, as shown in Supplementary Fig. 21. Comparisons of energy consumption with other artificial sensory spiking neurons are detailed in Supplementary Table 3. Further reductions in energy consumption are anticipated with improvements in $VO_2$ memristor technology, specifically by using a device with a lower $V_{th}$, reduced output current, and minimized parasitic capacitance.

## Flexible crossmodal in-sensor encoding and haptic-feedback system for human-machine interaction

The human proprioceptive reflex is essential for real-time processing of external perception information, which allows us to effectively manipulate object and avoid potential harming (Fig. 4a). Here, a bio-inspired neural reflex system is demonstrated to implement cross-modal in-sensor encoding and provide real-time haptic-feedback for human-machine interaction (Fig. 4b).

In this system, the CSSN synchronously encode pressure and temperature signals into neuronal spiking signals. We tested real-time neuronal spiking characteristics of the CSSN under various pressure and temperature stimuli (Fig. 4c). It can be found that the larger stimulus signals, i.e., pressure and temperature, results in a higher $f_{fir}$ and smaller amplitude. This phenomenon occurs due to the CSSN's dynamic response to multimodal inputs. Specifically, an increase in pressure generates a lower resistance at the pressure sensor, which is transduced into a higher $f_{fir}$. The amplitude of the output spikes is predominantly influenced by the $V_{th}$ of the $VO_2$-based memristor, which is inherently sensitive to thermal changes. As depicted in Supplementary Fig. 22a, as temperature rises, there is a corresponding decrease in the $V_{th}$ of the memristor. According to Eq. 1, this decrease in $V_{th}$ consequently results in a reduced $V_{out}$ (Supplementary Fig. 22b). The $f_{fir}$ information is used in the encoding and processing of the flexible hardware system. We systemically measured $f_{fir}$ of the CSSN under various pressures and temperatures (Fig. 4d). A significant increase in output $f_{fir}$ of the CSSN is realized as pressure and temperature increases. When applying pressure of 18 kPa and temperature

of 42 °C, the maximum $f_{fir}$ can reach 48.6 kHz. To further quantify the sensing intensity of $f_{fir}$, we set four levels to describe different feeling states as gentle touch (< 15 kHz), appropriate feeling (15–28 kHz), discomfort (28–40 kHz) and strong discomfort (> 40 kHz) (Fig. 4d). Once the uncomfortable threshold state ($f_{fri0}$ = 28 kHz) is reached, the CSSN immediately generates an uncomfortable sensing-feedback information as the biological reflex arc does. Notably, as the $V_{th}$ of the memristor decreases with increasing temperature, shifting the pressure threshold to a lower value which results in CSSN becoming sensitive. It is found that the pressure threshold of uncomfortable state linearly decreases with the temperature increasing, with detailed results shown in Supplementary Fig. 23. This means a low pressure can also cause uncomfortable feeling when the external temperature is very high. This phenomenon is similar to the "sensitization" of biological noci-ceptor neurons under injury, as elaborated in Supplementary Fig. 24 and Supplementary Note 3.

Based on the above CSSN, a flexible integrated in-sensor encoding and haptic-feedback system is demonstrated, as shown in Fig. 4e. This flexible hardware system is interfaced with a robotic hand through a WIFI module. Details of this system are explained in Supplementary Note 4. $f_m$ = 20 kHz and $f_{fri0}$ are set as the motion threshold and avoidance threshold for the robotic hand, respectively. This flexible hardware system is attached to the palm of the hand to perceive different stimuli and wirelessly transmitted to the robotic hand for remote interaction. Figure 4f and Supplementary Movie 1 show the dynamic responses of this system and the consequent haptic-feedback of the robotic hand under various environmental conditions. Initially, this flexible system is in the silent state without spiking response (Fig. f(i) and Supplementary Movie 1(i)). When a small ball (weak pressure) is placed in the human hand, the flexible system generates a neuronal spike with the $f_{fir1}$ (10.1 kHz). As the $f_{fir1}$ not reaching the $f_m$, the robotic hand does not receive a motion feedback signal, still remains at a silent state (Fig. f(ii) and Supplementary Movie 1(ii)). When human hand gripped the ball (strong pressure), this flexible system is stimulated to the appropriate feeling state, the output neuronal spike ($f_{fir2}$ = 22.3 kHz) exceeds $f_m$, resulting in motion feedback to make robotic hand to grasp (Fig. f(iii) and Supplementary Movie 1(iii)). When human hand grasped a hot water cup of ~45 °C (danger signal), this system can generate the motion feedback of open hand ($f_{fir3}$ = 31.4 kHz) to avoid injury (Fig. f(iv) and Supplementary Movie 1(iv)). These results demonstrate that the flexible spiking sensing-feedback hardware system can emulate human grasping and avoidance behaviors, showing its potential in human-machine interaction.

Some existing works reporting flexible sensing systems for haptic-feedback, are summarized in Supplementary Table 4. Some of these systems can only encode single-modal signals by using complex analog-to-digital converter (ADC) chips[36,50] or CMOS neuronal circuits[51]. Other systems with multimodal processing capabilities[4,52–54], can mimic biological multisensory pathway by using multiple sensors and external processing units (e.g., ADC), but cannot do in-sensor spike encoding. Compared with these systems, our flexible system is capable of synchronously in-sensor encoding bimodal signals and provides real-time haptic-feedback for human-machine interaction based on a compact CSSN circuit, without multiple sensors and complex conversion circuits. This system can dramatically simplify hardware complexity and reduce redundant data, which is crucial for constructing inexpensive and wearable electronics with low energy consumption.

## Crossmodal in-sensor spiking reservoir computing system

In addition to performing in-sensor encoding and haptic-feedback for human-machine interaction, the CSSNs are expected to possess accurate object recognition and feedback. We further applied CSSNs as sensory layer of the spiking reservoir network to construct a

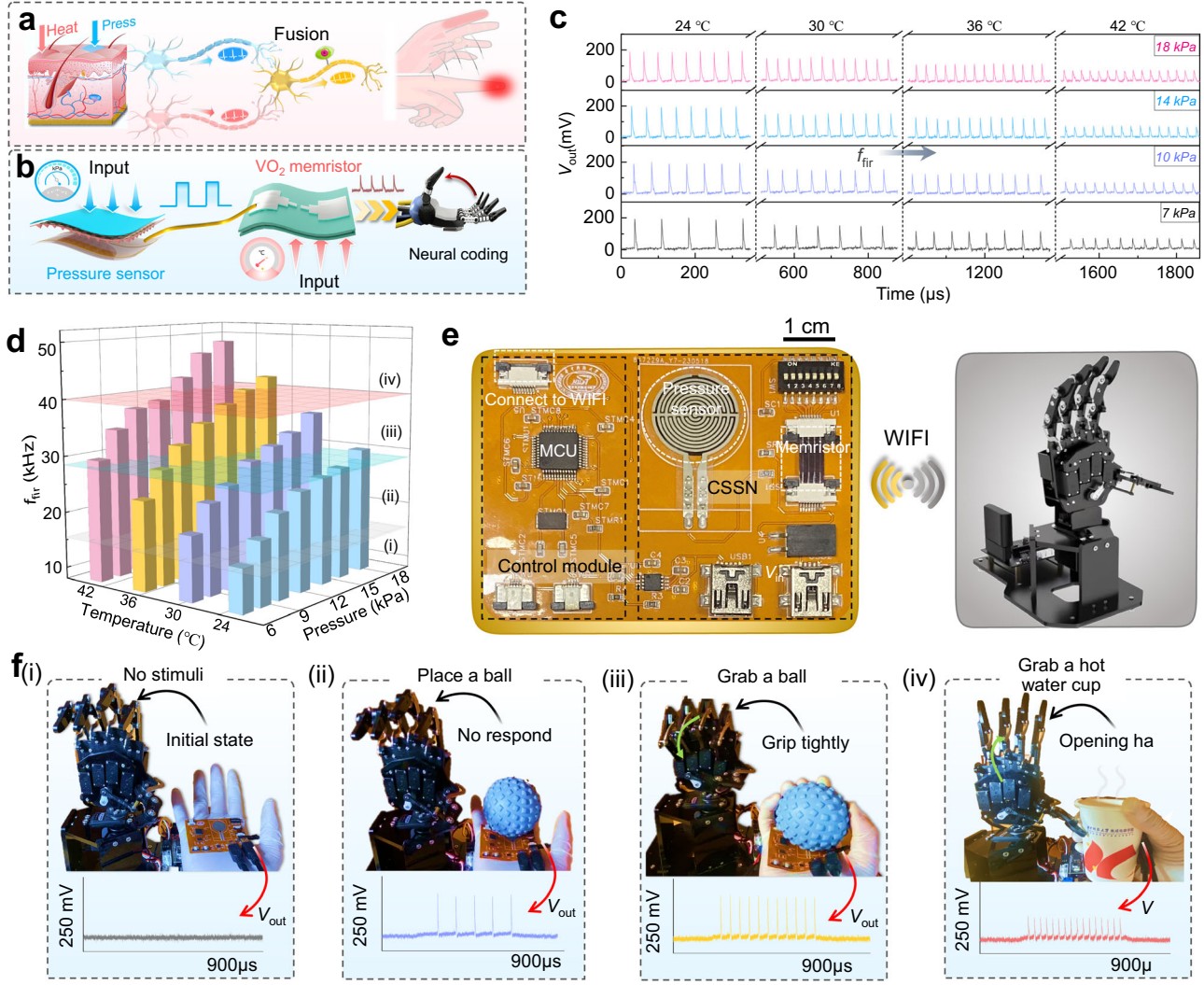

**Fig. 4 | Crossmodal in-sensor encoding and haptic-feedback for human-machine interaction. a** Schematic diagram of the biological multisensory feedback system. **b** Schematic diagram of the artificial CSSN for multimodal in-sensor neural encoding and feedback. **c** Real-time spiking responses of the CSSN under co-stimulation of pressure (7–18 kPa) and temperature (24–42 °C). **d** Output $f_{fri}$ of the CSSN to different pressures and temperatures. The spiking intensity of the $f_{fri}$ is divided into four sensation levels: (i) gentle touch; (ii) appropriate feeling; (iii) slightly discomfort; (iv) strong discomfort. **e** Photographs showing a flexible integrated sensing-feedback system (left) and a robotic hand (right). The CSSN encodes multimodal signals into spikes. The haptic-feedback of robotic hand was implemented by spike signals with different $f_{fir}$. **f** Real-time feedback signals of the robotic arm under different stimulus applied to human hand. Photos showing robotic arm making different gestures in response to different neuronal spikes correlating to different stimuli inputs: (i) no stimuli; (ii) place a ball on hand; (iii) grab a ball; (iv) grab a hot water cup.

crossmodal spiking reservoir computing system with in-sensor encoding capability (Figs. 5a, b, Methods and Supplementary Note 5). As is well known, spiking neural networks are difficult to train, especially when dealing with complex information. Interestingly, a spiking reservoir computing (i.e., liquid state machine) only needs to train the readout layer and is suitable for processing complex temporal/sequential information with the lowest training cost[55,56]. The in-sensor spiking reservoir computing system can directly encode and fuse multimodal information into spike sequences and pass them to the spiking reservoir layer for classification, thus further improving computational efficiency.

To verify the crossmodal object recognition capability of the proposed system, we collected the real-time dynamic tactile data of four objects, including "H", "U", "S", and "T"-shape, with different pressure and temperature information (Fig. 5a and Supplementary Fig. 26). Each of these pixels has a certain bimodal pressure-temperature information. The pressure and temperature stimuli are directly sensed and encoded by sensory neuron into the spike signals

of different $f_{fir}$ and amplitude, which are then fed into the spiking reservoir network for classification. Both $f_{fir}$ and amplitude are normalized to represent the various bimodal information effectively. To simulate the real environment, Gaussian noise is added to the dataset. We trained dataset for 500 training epochs. The dynamic object information can be divided into 8 different categories, including object shape ("H", "U", "S", and "T"-shape) with signal intensity (high and low). The unisensory and multisensory network architectures are shown in Supplementary Fig. 27. When the system is classifying the "H"-shape object, the spiking output of 8 LIF neurons are shown in Fig. 5c. From the output firing probability in the Fig. 5c, it can be seen that the system correctly identifies "H"-shape object (label 1) of a high input stimulus. The multisensory has the highest accuracy for recognition of 98.1% than pressure-only recognition of 83.6% and temperature-only recognition of 79.1% after 500 training epochs (Fig. 5d), proving the importance of crossmodal sensing and fusion. The detailed confusion matrix for the classification results is shown in Supplementary Fig. 28.

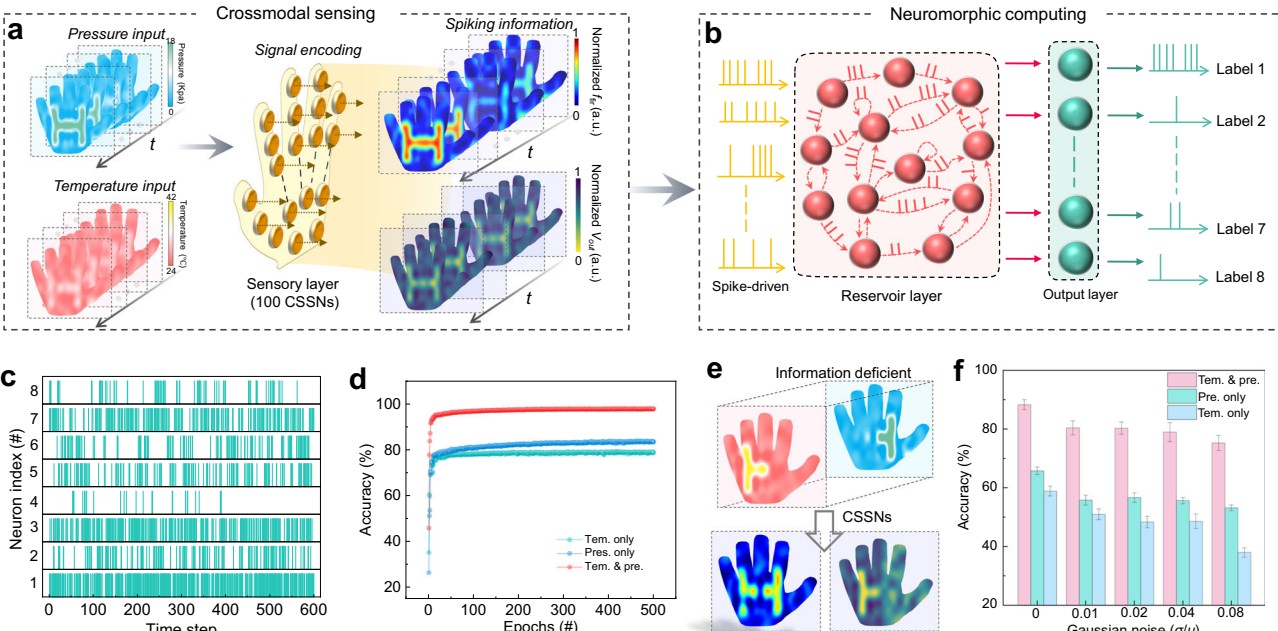

**Fig. 5 | Crossmodal in-sensor spiking reservoir computing for object recognition and feedback. a** Dynamic pressure (pre.) and temperature (tem.) information of objects is mapped to 100 CSSNs. The normalized mapping frequency and amplitude results are based on crossmodal (pre. and tem.) processing strategy. **b** The spiking reservoir network framework. **c** Spike firing of the output neurons. **d** The recognition accuracy for single-modal (sole pre. or tem.) and crossmodal processing strategies, respectively. **e** The crossmodal mapping with partial perceptual information loss. The right half hand can only sense pre., whereas the left half hand can only sense tem. **f** The recognition accuracy for the single-modal and crossmodal strategy. The accuracy was obtained by averaging the results of five tests. Crossmodal strategy showing the best accuracy when identifying objects with partial information loss.

Moreover, the challenges posed by partially damaged tactile data further highlight the efficacy of our multisensory fusion strategy. As shown in Fig. 5e, the pressure information of the "H"-shape object on the left half is lost and only be sensed via temperature sensory neurons, whereas the temperature information of this object in the right half is lacked and only be sensed via pressure sensory neurons. In such cases, reliance on a single-modal sensory layer results in substantially reduced accuracies, as low as 65.8% and 58.8% for temperature-only and pressure-only sensing, respectively. However, our multisensory fusion strategy effectively compensates for these deficiencies by integrating the available sensory information from the alternate modality. This approach enables the CSSNs in our sensory layer to reconstruct a fuller profile of the object, significantly enhancing the accuracy of object recognition to 88.3%, as shown in Fig. 5f. Furthermore, as the Gaussian noises increase, the performance of single-modal sensory systems (pressure-only or temperature-only) degrades markedly. In contrast, our bimodal pressure-temperature fusion strategy demonstrates robust capability by maintaining good recognition accuracy, preserving a rate of 75.2%. This distinct advantage of our crossmodal in-sensor spiking reservoir computing framework, especially in the multisensory fusion process, highlights its potential for deployment in environments with incomplete or corrupted sensory data.

## Discussion

Here, a flexible CSSN is realized based on a high-performance flexible $VO_2$ memristor for crossmodal in-sensor encoding. The memristor possesses electroforming-free TS behavior with high yield (97.8%), ultra-high endurance (> $10^{12}$), high C2C and D2D uniformity (0.72% and 3.73%, respectively), fast respond speed (< 30 ns), and excellent flexibility (bendable to a radius of 1 mm). The CSSN can synchronously encode temperature and pressure stimulus to spikes based on the coupled piezoresistive sensor and inherent temperature sensitivity of the $VO_2$, achieving crossmodal in-sensor encoding capability. A flexible hardware system based on CSSN is then designed, which can provide real-time haptic-feedback to external signals by encoding different spike events. A crossmodal in-sensor spiking reservoir computing system is demonstrated based on this CSSN, achieving recognition accuracy of 98.1% for dynamic objects and real-time sensing-feedback. The crossmodal in-sensor computing concept introduced here is a novel approach for sensing and processing multimodal signals directly within artificial neurons without complex data transmission. This simplifies the complexity of the hardware implementation and amplifies the functionalities for processing information at sensing terminals.

We have provided a comprehensive overview of recent advancements in artificial in-sensor spike encoding neurons[25,29,57–62], summarized in Table 1. Our CSSNs offer several advantages, including flexibility that support multimodal sensory inputs like tactile and temperature stimuli, emulating the human somatosensory system. Moreover, CSSNs demonstrate excellent endurance exceeding $10^{12}$ cycles and operate within an energy range of 3.9-50 nJ per spike. This positions them favorably in terms of energy efficiency and durability compared to other state-of-the-art in-sensor spike encoders. CSSNs support a wide range of applications, including dynamic object recognition and human-machine interactions. They are ideally suited for creating compact, versatile wearable sensory computing systems, potentially revolutionizing the landscape of portable and wearable technology. However, some challenges remain. The growth temperature of our $VO_2$ device is significantly lower than that of previously reported devices[23,29,32,33], further reduction while maintaining high performance is challenging. Despite the promising flexibility and stability of our flexible memristor, scaling CSSN array for mass integration without compromising performance is challenging. Additionally, managing the energy consumption of sensory neurons as the number of sensory nodes increases remains a challenge. There is substantial scope for enhancements through the co-optimization of materials, device architecture, and circuit design to overcome these limitations and fully realize the potential of the advanced sensory neurons.

**Table 1 | Comparison with reported in-sensor encoding neurons**

| Sensory signals | Crossmodal sensory | Sensory components | Flexibility | Endurance | Energy/spike | Coding-related applications | Ref. |
|---|---|---|---|---|---|---|---|
| Optical | No | 1 M | No | >100 | 2.1–20.3 nJ | Machine vision | 25 |
| Pressure &Temperature | Yes | 1 M + 1 PS | No | >10$^6$ | / | Object recognition | 29 |
| Optical | No | 1 M | No | >10$^4$ | ~190 nJ[a] | Image segmentation | 57 |
| Temperature | No | 1 M | No | >10$^3$ | ~0.15 nJ[a] | Edge detection | 58 |
| Physiological signals | No | 2 T | No | / | ~0.5 μJ[a] | Neuromorphic bio-interface | 59 |
| Optical | No | 1 M | No | >500 | ~32 pJ[a] | Pattern recognition | 60 |
| Pressure & optical | Yes | 1 T + 1 PS | No | / | ~8 nJ[a] | / | 61 |
| Optical | No | 1 T | No | >10$^8$ | ~0.1 nJ | Motion detection | 62 |
| Pressure &Temperature | Yes | 1 M + 1 PS | Yes | >10$^{12}$ | 3.9-50 nJ | Human-machine interaction & dynamic object recognition | This work |

[a]The energy consumption per spike is calculated approximately from the P–t, V–t and I–t curves in these reference papers, respectively.
To unify the benchmark, all the sensory components in these reference papers are equivalent to three categories: memristor (M), transistor (T), and pressure sensor (PS).

## Methods

### Device fabrication

The fabrication processes of flexible VO$_2$ memristors are briefed as follows. (1) To fabricate the smooth and thermostability (>300 °C) substrates, PI was spin-coated on top of clean Si/SiO$_2$ substrates. The coated substrates were cured in a nitrogen oven at 300 °C for 60 min. (2) The passivation layer of 500 nm SiO$_2$ was then evaporated onto the coated PI substrates by plasma enhanced chemical vapor deposition system (Oxford PlasmaPro 800 Stratum PECVD). (3) Cr$_2$O$_3$/VO$_2$ bilayer films were grown on the flexible substrates by magnetron sputtering (Denton Discovery 635) at 280 °C. VO$_2$ films were deposited using a V$_2$O$_3$ target (4 inches, 99.9% purity) with a 200 W DC power supply, maintaining pressure of 8 mTorr with 49.15 sccm Ar and 0.85 sccm O$_2$. The Cr$_2$O$_3$ layer was deposited using a Cr$_2$O$_3$ target (4 inches, 99.9% purity) with a 150 W RF power supply and 40 sccm Ar under pressure of 6 mTorr. (4) The planar electrodes consisting of Ti (10 nm)/Au (50 nm) were patterned on top of the films with electron-beam lithography (EBL, Vistec EBPG 5000 plus ES) along with electron beam evaporation and lift-off. The electrodes of 500-nm gap and 1-μm width were defined as shown in Fig. 2a, so that an electric field of the order of several tens of megavolts per meter could be generated by applying a few volts. (5) The SiO$_2$ covering layer was defined by UV photo-lithography and deposited by PECVD with a thickness of 100 nm. (6) The flexible VO$_2$ memristors were then peeled off from the Si substrate. The detailed fabrication process flow is shown in Supplementary Fig. 1.

### Device characterization

The surface morphology of the VO$_2$ films was examined by atomic force microscope (AFM, NT-MDT NTEGRA), X-ray diffraction (XRD, Bruker D8 DISCOVER) and Raman (Renishaw in Via). The Cross-sectional transmission electron microscopy (TEM) image and Energy dispersive X-Ray Spectroscopy (EDS) analysis were performed by the Talos F200X microscope. Measurements of conductive channels were performed by a commercial scattering-type scanning near-field optical microscopy (s-SNOM, Neaspec GmbH). The direct-current scanning measurements were conducted by a semiconductor analyzer (Agilent B1500A). Electrical pulse tests were carried out via the waveform generator/fast measurement unit (WGFMU) and semiconductor pulse generator unit (SPGU) of the semiconductor analyzer. For the tests of neuronal circuits, the pulse excitations applied across the circuits were generated by SPGU, while the output spikes were monitored using an oscilloscope (Keysight DSOX3104T) and a current waveform analyzer (Keysight CX3322A). The commercial flexible piezoresistive sensors (RF-C18.3-ST) were used in sensory neuron circuit. For the tests of the pressure sensors, the pressures were applied through the spring testing machine with a force gauge (ZQ-21B-1), and the electrical parameters were measured using an Agilent B1500A. For the heating experiments, a Lakeshore cryo-probe station was used for the temperature-dependent electrical characterizations. Before each measurement, the probe station was held at the set temperature for 4 mins to stabilize the device temperature.

### Neural network simulation

An in-sensor spiking reservoir computing system was implemented for multimodal object recognitions through simulation based on experimental data of the CSSNs. This network is composed of three layers: sensing layer for object perception, reservoir layer for data process, and readout layer for final learning classification, as shown in Fig. 5a, b. We developed a customized dataset including four types of real-time dynamic tactile data ("H", "U", "S", and "T"-shape objects, with different pressure and temperature information) for the spiking reservoir network training and recognition. Input dynamic pressure and temperature information are encoded into spiking signals (i.e., spiking frequencies and amplitudes) by a simulated array of 100 CSSNs, and then directly fed into the spiking reservoir network. The CSSN model, which is based on experimental data, employs interpolation to correlate spike amplitude and frequency information with perceived temperature and pressure, as shown in Supplementary Fig. 25. In the reservoir layers, there are 27 interacting virtual neurons. Simple artificial LIF neurons are used to construct the virtual neurons, where their parameters were extracted from experimental data of VO$_2$-based LIF neurons. The interaction between neurons is implemented by software. The readout layer is a 27 × 8 fully connected network, with eight label outputs representing the "H", "U", "S", and "T"-shape objects in sequence, each of which obtains shape information with input intensity (high or low). A supervised algorithm, stochastic gradient descent, is utilized to train the readout networks for the dynamic multimodal object recognition task. 600 samples are used for training set, and another 600 samples are used for testing set. The network is then implemented by python 3.9. More details are shown in Supplementary Note 5.

## Data availability

All data supporting this study and its findings are available within the article, its Supplementary Information and associated files. The source data have been deposited in the FigureShare database [https://figshare.com/s/61ebcbe49cc677cf0714].

## Code availability

The codes for the spiking reservoir computing are available from the corresponding author with detailed explanations upon request.

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

## Acknowledgements

This work was supported by the Natural Science Foundation for Distinguished Young Scholars of Hubei Province of China (Grant No.2023AFA065), the National Key Research and Development Program (Grant No. 2019YFB2205100), Hubei Province Key Scientific and Technological Project (Grant No. 2022AEA001), the National Natural Science Foundation of China (Grant No. 62175248), and Shanghai Science and Technology Funds (Grant No. 23ZR1481900).

## Author contributions

Z.Y.L., X.C., and R.Y. conceived the research. Z.Y.L., Z.S.L., and J.J.G. designed and fabricated the $VO_2$ devices and circuits. Z.Y.L., J.P.Y., L.S., and B.N.Z. carried out the electrical experiments. Z.Y.L., W.T., and J.P.Y. conducted the simulation. Z.Y.L., Y.F.L., and Z.P.D. carried out the device characterization tests. Z.Y.L., Y.X.D., J.X., P.J., and R.G.Y. assisted with data analysis and interpretation. Z.Y.L., R.Y., X.C., R.G.Y., and X.S.M. co-wrote the manuscript. All authors discussed the results and revised the manuscript.

## Competing interests

The authors declare no competing interests.

## Additional information

**Peerreview information** *Nature Communications* thanks Patrick (Jacob) Shamberger who co- reviewed with Rebeca Margarita Gurrola; Yuchao Yang and Jung Ho Yoon for their contribution to the peer review of this work. A peer review file is available.

