## [Peer Review File · Nature Communications]

REVIEWER COMMENTS

Reviewer #1 (Remarks to the Author):

Please see the attachment.

Reviewer #2 (Remarks to the Author):

Summary:

Manuscript # 492041 (“Crossmodal sensory neurons based on high-performance flexible memristors...”) describes a sensory network composed of a series of VO₂ based neurons which are grown on a flexible substrate, and which have been used to simultaneously sense temperature and pressure. This manuscript demonstrates a significant advance both in terms of the repeatability and reliability of the low-T array of VO₂ devices on a flexible substrate, as well as its implementation in a sensory feedback scheme. Based on the merits of these contributions, this is an important contribution and merits publication.

Overall, the manuscript is very well supported with sufficient supporting evidence and documentation. Data is analyzed and presented appropriately, although more supporting information on the reservoir computing scheme should be presented to make the reported recognition results to be more reproducible. Outside of this, I have a few recommendations that could make the manuscript more clear. Principally, these relate to the clarity of the presented interpretations, as well as clearly describing the experiment so that it can be repeated in other labs.

Major Points:

- 1) The reported increase in accuracy of the cross-modal function (to 98.1 recognition accuracy) is based on a reservoir computing scheme which is used to train 27 interacting virtual neurons. However, the description of this section is insufficient to be able to reproduce this process by other external readers. Please clarify the reservoir computing scheme that is used, including the description of the artificial LIF virtual neuron, and the nature and origin of the “600 samples” that are used for training and testing (e.g., what is considered a “sample” – is it data for some period of time under constant temperature and pressure conditions?) Are these actual data measured from your system? Purely synthetic data? Is any noise introduced?
- 2) The SNOM results illustrate some change in signal in the channel area. However, SNOM signal is complicated and is dependent on temperature as well as the local optical properties of the film. What does the illustrated signal change in the channel area represent? Is it simply localized heating? Does it imply a change in phase? Do you have standardized signal of the VO₂ film at different temperatures that can be used to interpret the signal between the electrodes?
- 3) The results of stability with cycling (up to 10¹² cycles) are fairly impressive. Please clarify – were these cycles completed by sweeping voltage (with an external compliance current of 1 mA)? Or were these cycles completed within an LIF circuit? If the former, are these results fully translatable to endurance in an LIF circuit?

Minor Points:

- 1) Please review the manuscript completely for syntax grammar and spelling. E.g., line 59 "... when a human touch an object, ..."; line 79 "in-senor spike..."; line 82, "Third, it is required the flexibility in-sensor..."; etc. throughout.
- 2) Please clarify the orientation of the bending relative to the devices.
- 3) Utilization of a 3D plot in figure 2f is not helpful, as it makes it difficult to discriminate the V_{th} at the different bending radii. I would recommend plotting this in a standard 2D plots with so the results at different bending radii can be directly compared.
- 4) The symbol illustrated in the LIF circuits (as well as the input in the video) implies that the externally applied voltage is applied in a pulse. Please clarify pulse length and duty factor of the input bias. What was the rationale for applying V_{in} in discrete pulses rather than in the constant steady state?
- 5) Supplementary figure 4 suggests a potential oxygen gradient in the VO₂ layer. Can you comment on this result and whether this reflects the existence of different oxidation states or is within instrumental resolution for the presented EDS results?
- 6) It would be helpful to clearly illustrate the impact of temperature on the response of the overall LIF circuit in fig. 3b (similar to that shown in fig. 3e for the effect of pressure). Specifically, it would be helpful at this point to clearly illustrate that temperature impacts both frequency and amplitude of the resulting spikes.
- 7) On Figure 3d, clearly illustrate the piezo-resistor element (R_p) in the circuit.
- 8) On suppl. Fig. 6 – you would not anticipate observing any peaks for VO₂(R) (e.g., Shvets et al 'A review of Raman spectroscopy of vanadium oxides' 2019). However, small peaks are observed in figure b and c at ~160 and ~300 C. Please comment on the origin and significance of these peaks. Does this imply phase impurity?
- 9) Similarly, in Suppl. Fig. 2 – there are minor XRD peaks at ~38, ~45 and ~57 deg 2theta. What are the origin of these peaks? Please index, and if possible, include a calculated powder diffraction pattern or PDF pattern for reference.
- 10) In Suppl. Fig. 5a, it is unclear what is the origin of the noise in the resistance signal at high temperatures (80 to 100 C). This appears to be more of a result of some electrical noise or issues with the device, rather than an attribute of the phase transition process itself (this is a very peculiar signal which is not commonly observed in VO₂ devices).
- 11) Specifics of the growth of the active VO₂ layer are not immediately clear. Under methods section in the main text, please expand upon (3) Cr₂O₃/VO₂ bilayer growth to at least include the type of target (V? VO₂?) and the partial pressure of the chamber during deposition.

Reviewer #3 (Remarks to the Author):

The manuscript titled "Crossmodal sensory neurons based on high-performance flexible memristors for humans-machine in-sensor computing system" reported flexible VO₂ memristors to realize a crossmodal in-sensor computing system for wearable human-machine interfaces. The fabricated system enables real-time processing of multimodal signals, achieving high accuracy in object identification and real-time signal feedback. While the work is of interest to the field, results are somewhat of an improvement on what was previously published, so this reviewer thinks the novelty part is lacking for publication in this journal. Also, the manuscript must be carefully improved considering below points.

1. In the manuscript, the author claims that the device has forming-free characteristics. However, there is no explanation as to why it has such characteristics. Additional explanation or analysis related to that must be provided.

2. It is known that materials-based threshold switching devices with MIT (Metal-Insulator Transition) characteristics, such as NbO₂, exhibit significant switching behavior variations depending on the active area size (the distance between in planar device). What is the active area size of the manufactured device or the distance between the electrodes of the planar device? To further validate the stability and scalability of the device, additional experiments should be conducted to observe how its properties evolve with changes in the active area size or the distance between electrodes.

3. What is the thickness of the buffer layer proposed in the manuscript, and is there any change in the crystallinity or device characteristics of the VO₂ thin film depending on the buffer layer thickness?

4. The author claimed that it mimics the human sensitization function by using the example of responding even when a small pressure is applied when exposed to high temperatures. However, Sensitization typically entails responding to the same non-harmful stimulus even upon sustained exposure. The author's claim about sensitization imitation should be presented in more detail or examined more closely.

5. Figure 4c shows a tendency for the maximum amplitude voltage of a spike to decrease as the stimulus (pressure) becomes stronger. As pressure increases, the voltage generated by the pressure sensor is expected to increase; thus, the applied voltage will also increase. Why does the absolute amplitude tend to decrease?

6. The proposed CSSN-based robot arm is very interesting due to its demonstration of the potential of device applications. However, there appears to be a deficiency in elucidating the device's novelty. The author proposed a comparison with various devices in the supplementary Table 1, but additional review is needed. The paper below, which implements multimodal sensing based on the same material (VO₂-based memristor) proposed, should be reviewed, and the novelty of the device should be additionally explained.

"Duan, Qingxi, et al. "Artificial multisensory neurons with fused haptic and temperature perception for multimodal in-sensor computing." *Advanced Intelligent Systems* 4.8 (2022): 2200039."

Reviewer #4 (Remarks to the Author):

Point-by-point response to reviewer:

Manuscript ID: NCOMMS-24-13684-T

Dear Reviewers,

We are grateful to the reviewers for providing the valuable and constructive comments to further improve the quality of our work. Considering the reviewers' evaluations, we have made a point-by-point response to the reviewers' comments, and revised our manuscript to improve the novelty and clarity of our work.

Please find below our responses (in **black**) to each of your specific comments (in *blue*). Revisions to the original article are indicated in **red**.

Reviewer #1 (Remarks to the Author):

This article by Li et al. reports on a bimodal sensory neuron fabricated on a flexible substrate, capable of transducing pressure and temperature information into spiking signals of varying frequency and amplitude. With this, a human-machine interaction system is demonstrated and a multi-modal object recognition task was simulated. The idea of combining an intrinsically temperature-sensing VO₂ spiking neuron and a separate sensor as the load resistor is not new, but realizing it on a flexible medium is interesting. In fact, the ability to grow high quality VO₂ thin films at low temperatures by introducing a Cr₂O₃ buffer layer to enable high device uniformity and reduced fabrication stress is meritorious, and could be something valuable beyond flexible electronics. However, there are some important issues that need to be addressed.

Response: Thank you very much for your affirmative comments on the innovative of our work and very constructive suggestions to improve our paper. According to your comments, more comprehensive experimental studies and detailed analyses have been conducted.

Our responses to your specific comments one by one are shown as follows.

Comment #1:

I suggest the authors compare the device variations with reported values in existing literature.

Response: Thank you very much for your valuable suggestion to benchmark our device variations against those reported in the existing literature.

In response to your suggestion, we compared our device with other reported threshold switching devices^{R1-R6} to highlight the improvements in device variations. As shown in Table R1, this analysis covers various parameters, including flexibility, forming characteristics, device-to-device (D2D) variations (key metrics such as the threshold voltage (V_{th}) distribution and its coefficient of variation (C_v)), and overall device yield. This detailed comparison clearly demonstrates the superior performance of our devices in terms of D2D variations and yield compared to other devices. This improvement is largely due to the high crystallinity of our VO₂ thin film achieved by introducing a Cr₂O₃ buffer layer, which ensures high yield and uniformity, and is essential for neural device applications. These attributes render our flexible neuronal device exceptionally robust and ideally suited for integration into advanced flexible sensory systems, establishing it as a key component poised to drive future technological advancements in this domain.

Device structure	Flexible	Forming-free	Device-to-device variation		Number of device (yield)	Ref.
			Distribution of V_{th}	$C_v (V_{th})$		
Ta/NbO _x /TiN	No	No	-1.57±0.17 V	6.51%	20	R1
Au/VO ₂ /Au	No	Yes	1.23±0.12 V	5.32%	10	R2
Pt/NbO _x /Pt	Yes	No	1.8±0.35 V	/	5	R3
Ag/OIHP/ITO/PEN	Yes	/	0.3±0.1 V	/	20	R4
Ag/FLBP-CsPbBr ₃ /ITO	Yes	/	1.15±0.46 V	19.74%	400 (84.3%)	R5
Ag/protein nanowire/Pt	Yes	No	65±14 mV	/	117	R6
Au/Cr ₂ O ₃ /VO ₂ /Au	Yes	Yes	1.84±0.11 V	3.73%	225 (97.8%)	This work

Table R1. Comparison of D2D variations in our flexible memristor with those reported in existing works.

Corresponding references:

- R1. Li, F. et al. A skin-inspired artificial mechanoreceptor for tactile enhancement and integration. *ACS Nano* **15**, 16422-16431 (2021).
- R2. Yuan, R. et al. A calibratable sensory neuron based on epitaxial VO₂ for spike-based neuromorphic multisensory system. *Nat. Commun.* **13**, 3973 (2022).
- R3. Ang, J. M., Dananjaya, P. A., Ang, C. C. I., Lim, G. J. & Lew, W. S. Strain-induced degradation and recovery of flexible NbO_x-based threshold switching device. *Sci. Rep.* **13**, 16000 (2023).
- R4. Tang, L. et al. Flexible threshold switching selectors with ultrahigh endurance based on halide perovskites. *Adv. Electron. Mater.* **8**, 2100771 (2021).
- R5. Wang, Y. et al. Memristor-based biomimetic compound eye for real-time collision detection. *Nat. Commun.* **12**, 5979 (2021).
- R6. Fu, T. et al. Self-sustained green neuromorphic interfaces. *Nat. Commun.* **12**, 3351 (2021).

To address this point, we have incorporated a comprehensive comparison of D2D variation within our manuscript.

Revised main text:

Page 7 (Line 181-183): “The comparisons of D2D variations with other reported TS devices are presented in Supplementary Table 2. The present device performs outstanding high yield and uniformity.”

Revised supplementary information:

The corresponding Table R1 have been appended as Supplementary Table 2 in the supplementary information.

Page 31 (Line 270-274): “This analysis encompasses various parameters, including D2D variations, highlighting crucial metrics such as the threshold voltage (V_{th}) distribution and its coefficient of variation (C_v), as well as overall device yield. This comprehensive comparison clearly demonstrates the superior performance of our flexible devices in terms of uniformity and reliability.”

Corresponding references R1-R6 have been added in the revised supplementary information.

Comment #2:

The mechanical and electrical endurance of the devices under bending can be due to the negligible geometry change of the active area given how small the devices are. Can the devices can also sustain other frequently encountered mechanical stresses such as stretching, which might affect the active area?

Response: Thank you very much for your valuable feedback. We agree that the small size of our devices likely contributes to their mechanical robustness under bending. To address the concern regarding the device's endurance to other mechanical stresses, such as stretching, which could affect the active area, we have conducted a thorough investigation. It is important to note that stretching introduces a distinct type of mechanical stress compared to bending and may present unique challenges.

We have performed systematic stretching tests to assess the mechanical endurance of our devices, as shown in Figure R1. Our findings indicate that as the tensile strain increases, the V_{th} and initial resistance (R_{off}) of the VO₂ devices gradually increase. This result is primarily attributed to the mechanical deformation of the VO₂ film and the electrodes. The substrate of our devices is the flexible polyimide (PI) film, which has a high Young's modulus (>2.5 GPa)^{R7}, demonstrating strong resistance to stretching. However, once stretched, the PI film cannot revert to its original state, leading to the pulling and eventual rupture of the VO₂ thin film and electrodes, affecting their switching performance. The VO₂ thin film and electrodes undergo microstructural degeneration, such as cracks (Figure R1(e)), which increase the resistance of the VO₂ thin film. The R_{off} increases due to the reduced conductive pathways within the deformed film. These changes require a higher voltage to achieve the switching behavior in the VO₂ memristor, resulting in an increased V_{th} . In practical wearable application scenarios, the stretching strain range of 0-10% and strong resistance to stretching that our PI-based devices can accommodate is sufficient for many flexible

device applications^{R8,R9}.

Notably, due to their high Young's modulus, PI substrates cannot revert to their initial state after stretching and are not suitable as stretchable substrates. For studying the stretchability of devices, it is preferable to select more elastic substrates such as polydimethylsiloxane (PDMS), polyurethane (PU), and Ecoflex^{R9}. These substrates require lower growth temperatures for the deposition of thin films to prevent thermal degradation. Specifically, PDMS substrates typically require a growth temperature below 200°C, while PU necessitate even lower temperatures, below 120°C. It is crucial to reduce the growth temperature of VO₂ to accommodate these flexible substrates. Future work should focus on further optimizing the experimental protocols to achieve lower growth temperatures for VO₂ film.

Figure R1. Characterization of the flexible VO₂ memristor under stretching conditions. **(a)** Optical images showing flexible VO₂ memristors under strains of 0% (initial), 3%, and 10% @ y direction. **(b-d)** *I-V* curves of the flexible VO₂ memristor at strains of 0-10%. **(e)** Surface morphology of the VO₂ film before and after stretching (10%).

Corresponding references:

- R7. Qu, C., Hu, J., Liu, X., Li, Z. & Ding, Y. Morphology and mechanical properties of polyimide films: The effects of UV irradiation on microscale surface. *Materials* **10**, 1329 (2017).
- R8. Liao, F. et al. Ultrasensitive flexible temperature-mechanical dual-parameter sensor based on vanadium dioxide films. *IEEE Electron Device Lett.* **38**, 1128-1131 (2017).
- R9. Lou, Z., Wang, L., Jiang, K., Wei, Z. & Shen, G. Reviews of wearable healthcare systems: Materials, devices and system integration. *Mater. Sci. Eng: R* **140**, 100523 (2020).

We have revised the manuscript to include these findings and to provide a more comprehensive discussion on the mechanical endurance of our devices under different types of mechanical stresses.

Revised main text:

Page 6 (Line 170-174): “The excellent robustness of flexible memristors under bending is primarily due to the negligible geometric change in the active area, facilitated by their small dimensions. Our stretching experiments revealed that these devices continue to function normally under tensile strains up to 10%, as shown in Supplementary Fig. 12.”

Revised supplementary information:

Figure R1 has been included in the supplementary information as Supplementary Fig. 12.

Page 13 (Line 138-143): “The flexible VO₂ memristors maintain switching behavior under tensile strains up to 10%. The V_{th} and initial resistance (R_{off}) of the flexible device increase with tensile strain due to permanent deformation and rupture of the VO₂ thin film and electrodes on the PI substrate. In practical wearable application scenarios, the stretching strain range of 0-10% and strong resistance to stretching that our PI-based VO₂ memristors can accommodate is sufficient for many flexible device applications^{4,5}.”

Corresponding references R8, R9 have been added in the revised supplementary information.

Comment #3:

The simple model describing the spiking behavior of the VO₂ devices provided is inadequate. I suggest the authors further provide the dependence of device parameters on sensory signals. Also, Eq. 2 and 3 are wrongly written. From Fig. 2c, I estimate the HRS and LRS to be ~6 kΩ and ~1.5 kΩ, and R_L is in the range of few kΩ (<16 kΩ for the piezoresistive sensor). The resistance of VO₂ cannot be neglected relative to R_L. Check Eq. 1 for mistakes, both R_{mem} and R_{on} refer to the resistance of VO₂, and R_{out} is not used.

Response: Thank you very much for your valuable comments. We sincerely apologize for not providing a detailed sensory neuron model in our previous manuscript. In response to your comments, we have extensively revised and supplemented our model and equations to enhance accuracy and comprehensiveness. We have developed a more detailed model of the VO₂ spiking sensory neuron (Equations R1-R12) to better capture the dynamics of the spiking behavior.

The core part of our model is an RC circuit governed by Kirchhoff's Current Law, which yields the differential equation:

$$C_m \frac{dV_m}{dt} + \frac{V_m}{R_{mem} + R_{out}} = \frac{V_{in} - V_m}{R_L} \quad (R1)$$

where C_m is the capacitance in parallel to the VO₂ device, R_{out} is the monitor resistor, and V_m represents the membrane potential. The resistance of VO₂ memristor is $R_{mem} = R_{on}$ in low resistance state (LRS) and $R_{mem} = R_{off}$ in high resistance state (HRS).

For the charging process where V_m rises from V_{hold} to V_{th} , we analyze the circuit at $R_{rise} = R_{off} + R_{out}$ and solve the differential equation with the initial condition $V_m(0) = V_{hold}$, leading to the following equation:

$$V_m(t) = V_{hold} \exp\left(\frac{-t}{r_H R_L C_m}\right) + V_{in} r_H \left(1 - \exp\left(\frac{-t}{r_H R_L C_m}\right)\right) \quad (R2)$$

where $r_H = \frac{R_{rise}}{R_L + R_{rise}}$ is the resistance factor when the VO₂ device is in the HRS.

At $t = t_{rise}$, $V_m(t_{rise}) = V_{th}$. Plugging these values into the equation and rearranging, we arrive at the expression:

$$t_{\text{rise}} = C_m (R_{\text{rise}} \parallel R_L) \times \ln \left(\frac{V_{\text{in}} r_H - V_{\text{hold}}}{V_{\text{in}} r_H - V_{\text{th}}} \right) \quad (\text{R3})$$

A similar analysis applies to the discharging process, where $R_{\text{fall}} = R_{\text{on}} + R_{\text{out}}$. By integrating the differential equation with initial condition $V_m(0) = V_{\text{th}}$, we obtain the following equation:

$$V_m(t) = V_{\text{th}} \exp\left(\frac{-t}{r_L R_L C_m}\right) + V_{\text{in}} r_L \left(1 - \exp\left(\frac{-t}{r_L R_L C_m}\right)\right) \quad (\text{R4})$$

where $r_L = \frac{R_{\text{fall}}}{R_L + R_{\text{fall}}}$ is the resistance factor when the VO₂ device is in the LRS.

At $t = t_{\text{fall}}$, $V_m(t_{\text{fall}}) = V_{\text{hold}}$, we obtain:

$$t_{\text{fall}} = C_m (R_{\text{fall}} \parallel R_L) \times \ln \left(\frac{V_{\text{in}} r_L - V_{\text{th}}}{V_{\text{in}} r_L - V_{\text{hold}}} \right) \quad (\text{R5})$$

Thus, oscillating frequency f_{fri} could be expressed as:

$$f_{\text{fri}} = \frac{1}{t_{\text{rise}} + t_{\text{fall}}} \quad (\text{R6})$$

This model is similar to the one given in references^{R10, R11}. Additionally, we have incorporated thermal effects into our model using Newton's Law of cooling^{R12}, leading to:

$$\frac{dT}{dt} = \frac{1}{C_{\text{th}}} \left(\frac{V^2}{R_{\text{Mem}}} - \frac{T - T_0}{R_{\text{th}}} \right) \quad (\text{R7})$$

where T , T_0 , V , C_{th} , and R_{th} are the actual temperature of the device, operating temperature, voltage across the VO₂ memristor, effective thermal capacitance, and thermal resistance, respectively. When the system reaches the steady state ($\frac{dT}{dt} = 0$), we obtain the following equation:

$$\frac{V^2}{R_{\text{mem}}} = \frac{T - T_0}{R_{\text{th}}} \quad (\text{R8})$$

When the device undergoes a resistance transition ($T = T_t$, where T_t is the transition temperature of the VO₂). By substituting $V = V_{\text{th}}$ and $V = V_{\text{hold}}$ into Equation R8, we derive:

$$V_{\text{th}}(T_0) = \sqrt{\frac{R_{\text{off}}}{R_{\text{th}}} (T_t - T_0)} \quad (\text{R9})$$

$$V_{\text{hold}}(T_0) = \sqrt{\frac{R_{\text{on}}}{R_{\text{th}}}} (T_{\text{t}} - T_0) \quad (\text{R10})$$

To extend the model to incorporate pressure influences, we introduce a new equation:

$$R_{\text{L}} = R_{\text{c}} + R_{\text{p}} \quad (\text{R11})$$

where R_{p} represents the resistance of pressure sensor. Substituting equation (R11) into equations (R3, R5, R6), we can analyze how changes in pressure affect the neuron's spiking behavior. Notably, when neurons encode both pressure and temperature information, we integrate the thermal model described by Equations (R9) and (R10) with the pressure model.

We sincerely apologize for the confusion caused by the previous usage. We have meticulously corrected this, ensuring that the spike amplitude (V_{out}) is now accurately expressed as follows:

$$V_{\text{out}} = V_{\text{th}} \times \frac{R_{\text{out}}}{R_{\text{out}} + R_{\text{on}}} \quad (\text{R12})$$

Corresponding references:

- R10. Wang, P., Khan, A. I. & Yu, S. Cryogenic behavior of NbO₂ based threshold switching devices as oscillation neurons. *Appl. Phys. Lett.* **116**, 162108 (2020).
- R11. Rosca, T., Qaderi, F. & Ionescu, A. M. High tuning range spiking 1R-1T VO₂ voltage-controlled oscillator for integrated RF and optical sensing. In *ESSDERC 2021 - IEEE 51st European Solid-State Device Research Conference (ESSDERC)* (IEEE, 2021).
- R12. Lee, S. B., Kim, K., Oh, J. S., Kahng, B. & Lee, J. S. Origin of variation in switching voltages in threshold-switching phenomena of VO₂ thin films. *Appl. Phys. Lett.* **102**, 063501 (2013).

Following the reviewer's comments, we have made the necessary corrections and added the following equations and discussions in the revised manuscript.

Revised main text:

Page 8 (Line 226-246): “According to the analysis of this circuit, the spike amplitude (V_{out}) is related to V_{th} , monitor resistor (R_{out}), and the LRS of the memristor (R_{on}) as

follows (Eq. 1):

$$V_{\text{out}}=V_{\text{th}}\times\frac{R_{\text{out}}}{R_{\text{out}}+R_{\text{on}}} \quad (1)$$

Thus, the amplitude of V_{out} mainly depends on V_{th} as the temperature increases. To study the dependence of f_{fri} on temperature, we first calculate the charging time (t_{rise}) and discharging time (t_{fall}) of the sensory neurons, which can be described by the following equations (Eqs. 2, 3):

$$t_{\text{rise}}=C_{\text{m}}(R_{\text{rise}}\parallel R_{\text{L}})\times\ln\left(\frac{V_{\text{in}}\frac{R_{\text{rise}}}{R_{\text{L}}+R_{\text{rise}}}-V_{\text{hold}}}{V_{\text{in}}\frac{R_{\text{rise}}}{R_{\text{L}}+R_{\text{rise}}}-V_{\text{th}}}\right) \quad (2)$$

$$t_{\text{fall}}=C_{\text{m}}(R_{\text{fall}}\parallel R_{\text{L}})\times\ln\left(\frac{V_{\text{in}}\frac{R_{\text{fall}}}{R_{\text{L}}+R_{\text{fall}}}-V_{\text{th}}}{V_{\text{in}}\frac{R_{\text{fall}}}{R_{\text{L}}+R_{\text{fall}}}-V_{\text{hold}}}\right) \quad (3)$$

where R_{rise} and R_{fall} are the $R_{\text{mem}}+R_{\text{out}}$ during the charging and discharging process, respectively. Furthermore, the f_{fri} can be expressed by Eq. 4:

$$f_{\text{fri}}=\frac{1}{t_{\text{rise}}+t_{\text{fall}}} \quad (4)$$

After considering the influence of temperature, the threshold and holding voltage can be described as follows⁴⁷:

$$V_{\text{th}}(T_0)=\sqrt{\frac{R_{\text{off}}}{R_{\text{th}}}}(T_{\text{t}}-T_0) \quad (5)$$

$$V_{\text{hold}}(T_0)=\sqrt{\frac{R_{\text{on}}}{R_{\text{th}}}}(T_{\text{t}}-T_0) \quad (6)$$

where R_{th} , T_{t} , and T_0 are the effective thermal resistance, the transition temperature of VO₂ and the operating temperature, respectively. Therefore, the impact of temperature on sensory neuron spiking can be obtained by inserting Eqs. (5) and (6) into Eqs. (1)-(4). For the temperature sensory neuron, the increase in f_{fri} mainly stems from the decrease in V_{th} and R_{mem} caused by rising temperature⁴⁸.

Page 10 (Line 270-278): “To further study the dependence of pressure on the f_{fri} , extending the model to pressure spiking sensory neuron, which can be described by the following equation:

$$R_L = R_c + R_p \quad (7)$$

The impact of pressure on sensory neuron spiking can be obtained by inserting Eq. (7) into Eqs. (1)-(4). Notably, when neurons encode both pressure and temperature information, the model needs to be built by combining Eqs. (5) and (6). For the pressure sensory neuron, the increase in f_{in} mainly stems from the decrease in R_p caused by rising pressure.”

Corresponding reference R12 has been added in the revised main text.

Comment #4:

In the cross-modal recognition task, I believe that the raw spikes from the CSSNs containing both frequency and amplitude information are used as the reservoir input, but both the main text and the Methods mentioned that only frequency is used. The authors should clarify this. Also, regarding the collected data, I believe that the temperature information and the pressure information are correlated, meaning that where the pressure is high (for example the strokes of the letters) the temperature is also high. Then, how does the network perform better on cross-modal data than on single-modal data when no new information is being presented to the network? What is the nature of the misclassified samples in the single-modal cases?

Response: Thank you very much for your insightful comments on our cross-modal recognition task. We appreciate the opportunity to clarify and expand on these points in our study. Here are the detailed responses to each point:

Regarding the input spike information used in our reservoir system, we primarily highlighted the use of frequency data from CSSNs because it is crucial for capturing the temporal dynamics essential for dynamic object recognition. CSSNs are indeed designed to encode both frequency and amplitude information based on dynamic pressure and temperature inputs. We apologize that our manuscript did not adequately emphasize this aspect. Therefore, we will revise it to explicitly detail how both frequency and amplitude contribute to our system’s input. Additionally, we have enhanced our CSSN model illustration to show the correlation between spike amplitude,

frequency, and the sensed temperature/pressure variables, thus providing a clearer picture of our data encoding strategy (see Figure R3). The frequency information is pivotal for identifying temporal patterns, while amplitude provides additional dimension by capturing the intensity of the stimuli. This enriches the encoding process and enhances the network’s ability to comprehensively integrate and process multisensory data.

Figure R3. The CSSN model, derived from experimental data, uses interpolation to establish a relationship between the spike amplitude **(a)**/frequency **(b)** and the perceived temperature and pressure.

On the performance of cross-modal vs. single-modal data, your observation about the correlation between temperature and pressure, where both tend to increase during object interactions such as the strokes of letters, is indeed insightful. While individual modalities might capture specific features of the data, their combination can provide complementary insights. For instance, pressure might highlight the force of the strokes, while temperature could capture the thermal dynamics. Together, they offer a richer and more detailed representation. Moreover, cross-modal data often contains redundant information that can reinforce the network’s understanding and improve its robustness. For example, the pressure and temperature modalities may both capture certain aspects of the strokes, but their combination allows the network to form a more comprehensive and nuanced representation.

Cross-modal data integration exploits this redundancy to refine and confirm predictions, enhancing the system’s robustness, particularly in conditions where one type of sensory

input is unreliable and noisy. Such capabilities are indispensable in scenarios where sensory inputs might be compromised. To demonstrate our system’s efficacy in handling incomplete sensory data, we have conducted tests simulating partial sensory loss. For example, in a scenario depicted in Figure R4, an “H”-shaped object lacked pressure information on one half and temperature information on the other. Single-modal systems struggled significantly in these tests, with recognition accuracies dropping to 65.8% and 58.8% for temperature-only and pressure-only sensing, respectively. In contrast, our multisensory fusion strategy effectively compensated for these deficits by integrating the available data from both modalities, achieving a significantly enhanced accuracy of 88.3%. This approach proved resilient even under increased Gaussian noise, maintaining a robust accuracy of 75.2%.

Additionally, our network handles the classification of stimulus intensity, assessing whether stimuli are excessively intense to preempt potential damage. This multimodal integration helps to mitigate erroneous responses in such evaluations, ensuring more reliable and accurate performance.

Figure R4. Crossmodal in-sensor spiking reservoir computing for object recognition in a complex environment. **(a)** Cross-modal mapping with partial perceptual information loss. The right half of the hand can only sense pressure, while the left half can only sense temperature. **(b)** Recognition accuracy for single-modal and cross-modal strategies, with the cross-modal strategy demonstrating superior accuracy in identifying objects despite partial information loss.

For misclassified samples in single-modal cases, misclassifications typically stem from the system’s inability to access comprehensive sensory information. For instance, when relying solely on temperature data, the system might fail to correctly identify objects with similar thermal properties but significantly different pressure characteristics. Similarly, relying exclusively on pressure data might not distinguish between objects with similar external pressures but different thermal responses. Additionally, some samples might be more susceptible to noise in a single modality, leading to misclassification.

Misclassifications are particularly frequent in situations involving partial sensory loss or varying modality intensities, where the absence of complementary sensory data from one modality leads to significant recognition errors. For example, if an object loses partial pressure modality information and only temperature modality information is available, and if the temperature information lacks sufficient resolution or features to compensate for the loss of pressure information, the system may fail to accurately identify the object’s shape and strength, leading to incorrect classification. As shown in Figure R4, cross-modal data helps to mitigate this issue by providing additional information, allowing the network to make more informed decisions.

To address these points, we have added the revised experimental results and following discussion into the revised manuscript.

Revised main text:

Page 14 (Line 393-395): “...the spike signals of different f_{fir} **and amplitude**, which are then fed into the spiking reservoir network for classification. **Both f_{fir} and amplitude are normalized to represent the various bimodal information effectively.**”

We revised the discussion on multisensory applications to highlight the comparative benefits of the cross-modal approach.

Page 15 (Line 408-425): “**Moreover, the challenges posed by partially damaged tactile data further highlight the efficacy of our multisensory fusion strategy.** As shown in Fig. 5e, the pressure information of the “H”-shape object on the left half is lost and only be

sensed via temperature sensory neurons, whereas the temperature information of this object in the right half is lacked and only be sensed via pressure sensory neurons. In such cases, reliance on a single-modal sensory layer results in substantially reduced accuracies, as low as 65.8% and 58.8% for temperature-only and pressure-only sensing, respectively. However, our multisensory fusion strategy effectively compensates for these deficiencies by integrating the available sensory information from the alternate modality. This approach enables the CSSNs in our sensory layer to reconstruct a fuller profile of the object, significantly enhancing the accuracy of object recognition to 88.3%, as shown in Fig. 5f. Furthermore, as the Gaussian noises increases, the performance of single-modal sensory systems (pressure-only or temperature-only) degrades markedly. In contrast, our bimodal pressure-temperature fusion strategy demonstrates robust capability by maintaining good recognition accuracy, preserving a rate of 75.2%. This distinct advantage of our crossmodal in-sensor spiking reservoir computing framework, especially in the multisensory fusion process, highlights its potential for deployment in environments with incomplete or corrupted sensory data.”

Fig. 5 | Crossmodal in-sensor spiking reservoir computing for object recognition and feedback. **a**, Dynamic pressure and temperature information of objects is mapped to 100 CSSNs. The normalized mapping frequency and amplitude results are based on crossmodal (pressure and temperature) processing strategy. **b**, The spiking reservoir network framework. **c**, Spike firing of the output neurons. **d**, The recognition accuracy

for single-modal (sole pressure or temperature) and crossmodal processing strategies, respectively. **e**, The crossmodal mapping with partial perceptual information loss. The right half hand can only sense pressure, whereas the left half hand can only sense temperature. **f**, The recognition accuracy for the single-modal and crossmodal strategy. Crossmodal strategy showing the best accuracy when identifying objects with partial information loss.

Page 19 (Line 519-523): “Input dynamic pressure and temperature information are encoded into spiking signals (i.e., spiking frequencies and amplitudes) by a simulated array of 100 CSSNs, and then directly fed into the spiking reservoir network. The CSSN model, which is based on experimental data, employs interpolation to correlate spike amplitude and frequency information with perceived temperature and pressure, as shown in Supplementary Fig. 25.”

Revised supplementary information:

The corresponding Figure R3 have been appended as **Supplementary Fig. 25**. The encoding amplitude results are added to **Supplementary Fig. 26**.

Supplementary Figure 26. (a-c) Dynamic encoding results of three objects (U, S, and T-shape) with different temperatures and pressures by the CSNNs. To demonstrate the dynamic pressing process, the pressure on each object is increased over

time. The temperature of each object is changed over time in the range of 21 to 42 °C.

Comment #5:

The short discussion on innocuous/noxious stimuli recognition seems unrelated to the cross-modal theme of the article and should be removed in my opinion.

Response: Thank you very much for your insightful feedback regarding the content of the manuscript.

The section discussing the recognition of innocuous and noxious stimuli was initially included to highlight the diverse capabilities and potential applications for human-computer interaction. However, upon reviewing your comment and reassessing the content, we understand your concern that this discussion may seem tangential or disconnected from the primary focus of cross-modal theme. We have removed the section on the recognition of harmless and harmful stimuli to make the content more closely aligned with the relevant context of the cross-modal theme.

To address this question, we have deleted the relevant information to the main text and the supplementary information as below.

Revised main text:

Delete page 16: ~~Furthermore, this crossmodal in sensor object recognition can help to process both innocuous and noxious stimuli, then provide real-time warning and prevent hardware damage caused by harsh environments where the detailed results and discussion are shown in Supplementary Fig. 21 and Supplementary Note 4.~~

Revised supplementary information:

Delete Supplementary Fig. 21 and Supplementary Note 4:

Supplementary Figure 21. Application of tactile feedback for innocuous and noxious signals. (a) Flow diagram showing the sensing-feedback strategy for robotic recognition. (b) Encoding results of CSNN for three different curvature objects (cube, sphere, and cone) with the same weight. Compared to pressing on square and spherical objects, pressing on conical objects are subjected to big pressure and have a high output frequency, indicating noxious signals. (c) The feedback accuracy of the robotic arm while the robotic hand contacts different curvature objects. Detailed discussions are shown in Supplementary Note 4.

Supplementary Note 4: Crossmodal in-sensor spiking reservoir computing system for real-time feedback.

The sensing devices might suffer partial malfunctions due to their contact with noxious stimuli. The recognition of harmful signals can provide timely feedback and effectively prevent damage of hardware system. To prove the proposed system's capability to process innocuous and noxious signals, objects with three curvatures were chosen to simulate the scenarios of grasping objects by robots (Supplementary Fig. 21a, b). Specifically, “cone” shape object simulates the emergent situation of a sharp pressure, while “sphere” and “cube” shape objects simulate the normal situation of a uniform pressure. Based on the sensing-computing system, the high-intensity and low-intensity stimulus classification of object reflects the noxious and innocuous of input signal, respectively. The robotic hand had a high likelihood of sensing harm feedback while grasping cone (92.4%) and a significantly low likelihood of sensing harm for sphere

~~(18.0%) and cube (18.9%), see (see Supplementary Fig. 21c). Such ability to process both innocuous and noxious stimuli could provide real-time warning and prevent hardware damage caused by harsh environments.~~

Comment #6:

Achieving low power consumption is very important in areas where CSSNs are to be used in. I suggest the authors provide some power consumption figures and relevant comparisons.

Response: Thank you very much for your valuable feedback, and we sincerely apologize for the oversight in presenting power consumption data and comparisons in our initial submission. In response to your feedback, we have conducted additional measurements of power consumption for each spike signal in our CSSNs and compared these results with relevant state-of-the-art devices^{R2,R5,R6,R13-R15}.

The transient power was calculated by multiplying the input voltage with the output current. We then calculated the energy consumption per spike by dividing the total energy consumption by the number of spikes. As shown in Figure R5, the energy consumption for each spike was calculated as ~50 nJ. By leveraging the parasitic capacitance of the VO₂ memristor, we managed to reduce the energy consumption to as low as ~3.9 nJ per spike event.

Figure R5. The energy consumption of the CSSN for each spike event, depicting variations under different capacitance scenarios.

As suggested by the reviewer, we have included comparisons with relevant state-of-

the-art devices as sensory neurons to demonstrate the improvements. As shown in Table R2, CSSN offer significance advantages, such as flexibility, crossmodal sensory, multimodal in-sensor coding, fast response, and low power consumption. These attributes are crucial for energy-efficient wearable human-machine interfaces.

Although the energy consumption of our CSSN is not the lowest recorded, we are optimistic about achieving further reductions. Potential improvements could include using VO₂ memristors with a lower V_{th} , smaller output currents, and reduced parasitic capacitance.

Neuron device	Flexibility	Crossmodal sensory	Sensory signal	Highest spike frequency	Energy/spike	Ref.
Au/VO ₂ /Au memristor	No	No	Pressure, optical, curvature, temperature (near-sensor)	1.3 MHz	2.9 nJ	R2
Ag/FLBP-CsPbBr ₂ /ITO memristor	Yes	No	Optical (in-sensor)	25 Hz	2.1-20.3 nJ	R5
Ag/protein nanowire/Pt memristor	Yes	Yes	Optical, pressure, humidity (near-sensor)	0.25 Hz	~14 nJ*	R6
Si/NbO _x /TiN memristor	No	No	Pressure (near-sensor)	1.1 MHz	38 pJ	R13
Pt/V ₂ O ₅ /Pt memristor	No	No	Optical (in-sensor)	82 kHz	~190 nJ*	R14
Ag/TaO _x /ITO memristor	No	No	Optical (near-sensor)	200 Hz	~2.5 nJ*	R15
Ag/TaO _x /AlO _x /ITO memristor	No	No	Temperature (in-sensor)	1.6 kHz	~0.15 nJ*	R16
Organic electrochemical transistor	No	No	Physiological signals (in-sensor)	55 Hz	~0.5 μJ*	R17
Au/Cr ₂ O ₃ /VO ₂ /Au memristor	Yes	Yes	Pressure, temperature (in-sensor)	650 kHz	3.9-50 nJ	This work

*The energy consumption per spike is calculated approximately from the P-t, V-t and I-t curves in these reference papers, respectively.

Table R2. Comparisons with other state-of-the-art memristor-based sensory spiking neuron. To demonstrate the high-efficiency coding capabilities of the CSSN as a sensory neuron, we compared several key characteristics of artificial sensory neurons. These characteristics include the type of neuron device, flexibility, crossmodal sensory, sensory modality, highest spike frequency, and energy per spike.

Corresponding references:

- R13. Zhang, X. et al. An artificial spiking afferent nerve based on mott memristors for neurorobotics. *Nat. Commun.* **11**, 51 (2020).
- R14. Nath, S. K. et al. Optically tunable electrical oscillations in oxide-based memristors for neuromorphic computing. *Adv. Mater.* 2400904 (2024).
- R15. Chen, C. et al. A photoelectric spiking neuron for visual depth perception. *Adv. Mater.* **34**, 2201895 (2022).

- R16. Shi, K. et al. An oxide based spiking thermoreceptor for low-power thermography edge detection. *IEEE Electron Device Lett.* **43**, 2196-2199 (2022).
- R17. Sarkar, T. et al. An organic artificial spiking neuron for in situ neuromorphic sensing and biointerfacing. *Nat. Electron.* **5**, 774-783 (2022).

To comprehensively address your point, we have added these findings and discussions to the main text.

Revised main text:

Page 11 (Line 296-303): “High energy efficiency is critical in artificial sensory systems. To validate the efficiency of CSSNs in constructing high-efficiency human-machine interfaces, we further investigated its energy consumption per spike. We achieved a minimum of ~3.9 nJ per event with optimal parasitic capacitance settings, as shown in Supplementary Fig. 21. Comparisons of energy consumption with other artificial sensory spiking neurons are detailed in Supplementary Table 3. Further reductions in energy consumption are anticipated with improvements in VO₂ memristor technology, specifically by using a device with a lower V_{th} , reduced output current, and minimized parasitic capacitance.”

Revised supplementary information:

Page 22 (Line 214-217): “The transient power is calculated by multiplication of input voltage with output current, and the energy consumption for each spike is calculated by dividing the total energy consumption by the spike number.”

Figure R5 and Table R2 have been included in the supplementary information as Supplementary Fig. 21 and Supplementary Table 3, respectively. Corresponding references R2, R5, R6, R13-R17 have also been added in the revised supplementary information.

Comment #7:

While the authors did compare their work with some existing flexible haptic-feedback systems, I suggest the authors also include works on other flexible in-sensor encoders

to make the comparison more comprehensive. Also, it would be good to also compare the performance/efficiency of the proposed system with other systems on conventional rigid substrates, both single-modal and multi-modal. A brief discussion on limitations/challenges still currently faced by the flexible CSSN will very much benefit the readers.

Response: Thank you for your constructive feedback, which has provided a valuable opportunity to enhance the depth and breadth of our manuscript. In accordance with your suggestions, we have significantly expanded our comparative analysis to include a wider range of in-sensor spike encoding neurons and have detailed the challenges and limitations of our flexible CSSNs.

Expanded comparative analysis: We have broadened our comparison to include both flexible and rigid substrate-based in-sensor spike encoding neurons^{R5, R12, R15-R20}, covering both single-modal and multi-modal configurations, as depicted in Table R3. This comprehensive comparative analysis now highlights the distinguishing features of various state-of-the-art in-sensor encoders, such as sensory signals, components, flexibility, endurance, energy efficiency, and specific applications.

Generally, most of the state-of-the-art in-sensor spike sensory neuron are single-modal in-sensor spike encoder inspired by one of the human senses, as demonstrated in the table below. Additionally, these encoders use memristors/transistors on rigid substrates, making it difficult to construct flexible integrated sensory-computing systems. These encoders demonstrate outstanding performance in terms of energy efficiency and diverse applications. Potential applications predominantly include edge detection, image segmentation, pattern recognition, neuromorphic bio-interfaces, motion detection, and machine vision.

In contrast, our crossmodal in-sensor spike neuron exhibits several important features distinct from recently reported ones. First, unlike typical single-modal systems that mimic a single human sense, our system integrates tactile and temperature stimuli, offering richer and more diverse sensory input akin to the human somatosensory system. Second, our flexible crossmodal encoder design adopts a compact and flexible 1M+1PS

architecture, with the flexible memristor achievable through large-scale and scalable fabrication techniques. This enables the creation of portable and wearable sensory computing systems. Third, our encoder achieves endurance exceeding 10^{12} cycles and operates within an energy range of 3.9-50 nJ per spike, highly competitive within the field. These features contribute to high stability and low power consumption, facilitating the development of energy-efficient sensory computing systems. Fourth, our encoder's broad application potential ranges from human-machine interaction to dynamic object recognition, crucial for advancing tactile intelligence and damage prevention in sensory robotics and smart interfaces. We hope this pioneering effort will pave the way for future advancements in the field. Therefore, our work significantly differentiates from most existing works in this field.

Limitations and challenges: We acknowledge several challenges and limitations that need addressing as our technology advances. First, the growth temperature of our VO₂ device is significantly lower than that of previously reported VO₂ devices^{R2,R19,R22,R23}. However, further reducing the growth temperature while maintaining high performance to accommodate various application scenarios, such as stretchable electronic devices, remains a challenge. Second, while our flexible memristor shows promising flexibility and stability, scaling CSSN array for mass integration without compromising performance is challenging. Third, although our system is designed for energy efficiency, the relative high energy consumption per spike, compared to some rigid systems, indicates room for improvement. We are actively working on optimizing the growth conditions of the VO₂ film to increase the device resistance, thereby reducing power consumption. Scaling this efficiency to systems with thousands of sensory nodes requires innovative solutions in power management and circuit design, which we are currently developing.

Sensory signals	Crossmodal sensory	Sensory components	Flexibility	Endurance	Energy/spike	Coding-related applications	Ref.
Optical	No	1M	No	>100	2.1-20.3 nJ	Machine vision	R5
Optical	No	1M	No	>10 ⁴	~190 nJ*	Image segmentation	R14
Temperature	No	1M	No	>10 ³	~0.15 nJ*	Edge detection	R16
Physiological signals	No	2T	No	/	~0.5 μ J*	Neuromorphic bio-interface	R17
Optical	No	1M	No	>500	~32 pJ*	Pattern recognition	R18
Pressure & Temperature	Yes	1M+1PS	No	>10 ⁶	/	Object recognition	R19
Pressure & optical	Yes	1T+1PS	No	/	~8 nJ*	/	R20
Optical	No	1M+1T	No	>10 ⁸	~0.1 nJ	Motion detection	R21
Pressure & Temperature	Yes	1M+1PS	Yes	>10 ¹²	3.9-50 nJ	Human-machine interaction & dynamic object recognition	This work

*The energy consumption per spike is calculated approximately from the P-t, V-t and I-t curves in these reference papers, respectively. To unify the benchmark, all the sensory components in these reference papers are equivalent to three categories: memristor (M), transistor (T), and pressure sensor (PS).

Table R3. Comparison with the state-of-the-art in-sensor encoding neurons.

Corresponding references:

- R18. Wang, X., et al. Vertically integrated spiking cone photoreceptor arrays for color perception. *Nat. Commun.* **14**, 3444 (2023).
- R19. Duan, Q. et al. Artificial multisensory neurons with fused haptic and temperature perception for multimodal in-sensor computing. *Adv. Intell. Syst.* **4**, 2200039 (2022).
- R20. Sadaf, M. U. K., Sakib, N. U., Pannone, A., Ravichandran, H. & Das, S. A bio-inspired visuotactile neuron for multisensory integration. *Nat. Commun.* **14**, 5729 (2023).
- R21. Wang, R. et al. 1-phototransistor-1-threshold switching optoelectronic neuron for in-sensor compression via spiking neuron network. In *2023 International Electron Devices Meeting (IEDM)* 1-4 (IEEE, 2023).
- R22. Li, G. et al. Photo-induced non-volatile VO₂ phase transition for neuromorphic ultraviolet sensors. *Nat. Commun.* **13**, 1729 (2022).
- R23. Deng, S. et al. Selective area doping for Mott neuromorphic electronics. *Sci. Adv.* **9**, eade4838 (2023).

Following the reviewer's comments, we have created a table (Table 1) comparing the results of various in-sensor spike encoder. And, we added a section discussing the

comparison and challenges associated with our flexible CSSN in the revised manuscript.

Revised main text:

Page 16 (Line 448-467): “We have provided a comprehensive overview of recent advancements in artificial in-sensor spike encoding neurons^{25,29,57-62}, summarized in Table 1. Our CSSNs offer several advantages, including flexibility that support multimodal sensory inputs like tactile and temperature stimuli, emulating the human somatosensory system. Moreover, CSSNs demonstrate excellent endurance exceeding 10^{12} cycles and operate within an energy range of 3.9-50 nJ per spike. This positions them favorably in terms of energy efficiency and durability compared to other state-of-the-art in-sensor spike encoders. CSSNs support a wide range of applications, including dynamic object recognition and human-machine interactions. They are ideally suited for creating compact, versatile wearable sensory computing systems, potentially revolutionizing the landscape of portable and wearable technology. However, some challenges remain. The growth temperature of our VO₂ device is significantly lower than that of previously reported devices^{23,29,32,33}, further reduction while maintaining high performance is challenging. Despite the promising flexibility and stability of our flexible memristor, scaling CSSN array for mass integration without compromising performance is challenging. Additionally, managing the energy consumption of sensory neurons as the number of sensory nodes increases remains a challenge. There is substantial scope for enhancements through the co-optimization of materials, device architecture, and circuit design to overcome these limitations and fully realize the potential of the advanced sensory neurons.”

Corresponding references R5, R14, R16-R23 have been added in the revised main text.

Other issues

Comment #8:

Inconsistent usage of R_{mem} and R_{men} .

Response:

Thank you for your meticulous review and for identifying the errors in our subscript notation. Really sorry about the faults for our subscript notation. We have reviewed the entire manuscript to ensure that the representation of the memristor resistance (R_{mem}) are consistent. As we have rederived the formula for the sensory neuron model, it has now been removed from the manuscript.

Comment #9:

Line 87: Typo in “in-senor”, please check others carefully.

Response: Thank you for your attention to detail and for pointing out the spelling errors in our manuscript. Following your suggestion, we have thoroughly reviewed and polished the text to ensure a smoother reading experience for all.

Revised main text:

Page 3 (Line 79): “Second, it is difficult to achieve crossmodal **in-sensor**...”

Page 4 (Line 87): “The CSSNs perform **in-sensor** spike encoding...”

Page 13 (Line 368): “our flexible system is capable of synchronously **in-sensor** encoding...”

Comment #10:

Fig. 3d caption: Should “ R_l ” be “ R_c ” instead?

Response:

Thank you for identifying the errors in our subscript notation. We have now corrected it in the revised manuscript.

Revised main text:

Page 28 (Line 808): “ $R_c/R_{\text{out}} = 3 \text{ k}\Omega/50 \text{ }\Omega$ were used in the circuit.”

Comment #11:

Misaligned vertical axis in Fig. 5c

Response: Thank you for pointing out the error in the vertical axis of our Fig. 5c. We have redrawn the graph in the revised manuscript.

Revised main text (Page 30):

Reviewer #2 (Remarks to the Author):

Summary:

Manuscript # 492041 (“Crossmodal sensory neurons based on high-performance flexible memristors...”) describes a sensory network composed of a series of VO₂ based neurons which are grown on a flexible substrate, and which have been used to simultaneously sense temperature and pressure. This manuscript demonstrates a significant advance both in terms of the repeatability and reliability of the low-T array of VO₂ devices on a flexible substrate, as well as its implementation in a sensory feedback scheme. Based on the merits of these contributions, this is an important contribution and merits publication.

Overall, the manuscript is very well supported with sufficient supporting evidence and documentation. Data is analyzed and presented appropriately, although more supporting information on the reservoir computing scheme should be presented to make the reported recognition results to be more reproducible. Outside of this, I have a few recommendations that could make the manuscript more clear. Principally, these relate to the clarity of the presented interpretations, as well as clearly describing the experiment so that it can be repeated in other labs.

Response: Thank you very much for commenting the significance of our work and giving us insightful suggestions to further improve the quality of this work. Based on your comments, we have performed more thorough testing and characterization of the device, and supplemented the discussion with a more detailed explanation of the reservoir computing scheme to make the reported results to be more reproducible.

Our responses to your specific comments one by one are shown as follows.

Major Points:

Comment #1:

The reported increase in accuracy of the cross-modal function (to 98.1 recognition accuracy) is based on a reservoir computing scheme which is used to train 27 interacting virtual neurons. However, the description of this section is insufficient to be

able to reproduce this process by other external readers. Please clarify the reservoir computing scheme that is used, including the description of the artificial LIF virtual neuron, and the nature and origin of the “600 samples” that are used for training and testing (e.g., what is considered a “sample” – is it data for some period of time under constant temperature and pressure conditions?) Are these actual data measured from your system? Purely synthetic data? Is any noise introduced?

Response: Thank you very much for your detailed feedback. We appreciate the opportunity to clarify and expand upon the reservoir computing scheme and related aspects to ensure reproducibility for external readers.

For reservoir computing scheme, our crossmodal in-sensor spiking reservoir computing system, developed in Python 3.9, is designed with three layers: sensory layer, reservoir layer, and readout layer. The sensory layer employs an array of 100 crossmodal spiking sensory neurons (CSSNs) to convert dynamic pressure and temperature inputs into spiking amplitude and frequency signals. The CSSN model, derived from experimental data, uses interpolation to establish a relationship between the spike amplitude/frequency and the perceived temperature and pressure, as shown in Figure R1. The reservoir layer of our network serves as a dynamic processing unit that exploits the nonlinear capabilities of its architecture to extract essential features from input spike signals. This layer consists of 27 LIF neurons based on our VO₂ devices. Connectivity among neurons is randomly established, ensuring a dynamic and rich interaction pattern conducive to complex information processing. The behavior of these neurons follows these equations:

$$u_o(t) = u_o(t - \Delta t) \exp\left(-\frac{\Delta t}{RC}\right) + U(t) \quad (\text{R1})$$

$$s_o(t) = \mathcal{G}(u_o(t)) = \begin{cases} 1, & u_o(t) \geq V_{\text{th}} \\ 0, & u_o(t) < V_{\text{th}} \end{cases} \quad (\text{R2})$$

$$u_o(t) = \begin{cases} 0, & \text{if } s_o(t) \\ u_o(t), & \text{else} \end{cases} \quad (\text{R3})$$

These equations represent the leaky integration, firing, and resetting behaviors of the neurons, providing a framework for temporal integration and spike generation based on the synaptic inputs received from the sensory layer, which encodes real-time

temperature and pressure data into spikes. In our system, the threshold voltage V_{th} is set to the firing voltage of our neuronal circuit at 0.18 V, and the leaky constant (RC) is 20 μ s. The readout layer consists of a fully connected network of 27 neurons linked to eight output labels. These labels correspond to four different object shapes, each represented under high and low input intensities. This configuration allows for effective shape recognition and enables the network to differentiate between varying signal strengths, providing detailed feedback on object manipulation.

Figure R1. The CSSN model, derived from experimental data, uses interpolation to establish a relationship between the spike amplitude **(a)**/frequency **(b)** and the perceived temperature and pressure.

For dataset description, our dataset consists of 1,200 samples, with each sample representing a sequence capturing the dynamic interaction of a human hand pressing objects shaped as “H”, “U”, “S”, and “T”. The data for each pressing action includes 100 distinct sensory pixels, capturing specific moments across the interaction period. Gaussian noise is introduced to mimic real-world unpredictability and to test the system’s robustness. Noise levels are varied systematically across the dataset ($\delta/\mu = 0, 0.01, 0.02, 0.04, 0.08$) to assess performance under different signal-to-noise ratios. Each sample records the gradual changes in temperature and pressure during a press, encoded into spike frequency by a hand sensory array consisting of 100 CSSNs. The sensory array converts the physical stimuli into spike signals that are then processed by the reservoir layer.

The dataset is synthetic, generated based on a detailed simulation of tactile interactions

informed by experimental data, to closely replicate the real-world dynamics of object manipulation. To validate the efficacy of the learning algorithm and the generalizability of the neural network, the dataset is divided into training and testing subsets, with 600 samples each. This division ensures that the system can be rigorously evaluated on both known and unseen data, providing a robust assessment of its capabilities.

To address this question, we further explain the network simulation details, as shown below.

Revised main text:

Page 14 (Line 378-380): “We further applied CSSNs as sensory layer of the spiking reservoir network to construct a crossmodal spiking reservoir computing system with in-sensor encoding capability (Fig. 5a, b, Methods and Supplementary Note 5).”

Page 19 (Line 518-523): “Input dynamic pressure and temperature information are encoded into spiking signals (i.e., spiking frequencies and amplitudes) by a simulated array of 100 CSSNs, and then directly fed into the spiking reservoir network. The CSSN model, which is based on experimental data, employs interpolation to correlate spike amplitude and frequency information with perceived temperature and pressure, as shown in Supplementary Fig. 25.”

Revised supplementary information:

Figure R1 has been included in the supplementary information as Supplementary Fig. 25.

Supplementary Note 5: Simulation details about the crossmodal in-sensor spiking reservoir computing system simulations.

We developed a custom dataset comprising 1,200 samples for training and testing a neural network, designed to replicate the tactile interaction of a human hand pressing objects shaped as “H”, “U”, “S”, and “T”. Each sample captures a pressing action with gradual increases in temperature and pressure, consisting of 100 distinct sensory pixels. These pixels record specific instances within the pressing sequence along with corresponding temperature and pressure data over time. To mimic real-world

conditions and assess network robustness, we systematically introduced Gaussian noise of varying intensities ($\delta/\mu = 0, 0.01, 0.02, 0.04, 0.08$) across the dataset. The dataset was split equally into 600 samples for training and 600 for testing.

We implemented a crossmodal in-sensor spiking reservoir computing system in Python 3.9, comprising a sensory layer, a reservoir layer, and a readout layer based on experimental results. In the sensory layer, a 100-node CSSN array simulates the human hand, encoding dynamic pressure and temperature into spike amplitude and frequency signals. The reservoir layer consists of 27 LIF neurons. Connectivity among neurons is randomly established, ensuring a dynamic and rich interaction pattern conducive to complex information processing. Neurons receive spike signals from CSSN nodes, with random synaptic connections promoting diverse neural interactions. Neuronal connectivity follows geometric rules, with synaptic weights uniformly set at 12 for consistent signal processing. The membrane potential dynamics of LIF neurons are described by the equations:

$$u_o(t) = u_o(t-\Delta t) \exp\left(-\frac{\Delta t}{RC}\right) + U(t) \quad (1)$$

$$s_o(t) = \mathcal{G}(u_o(t)) = \begin{cases} 1, & u_o(t) \geq V_{th} \\ 0, & u_o(t) < V_{th} \end{cases} \quad (2)$$

$$u_o(t) = \begin{cases} 0, & \text{if } s_o(t) \\ u_o(t), & \text{else} \end{cases} \quad (3)$$

The first term in equation (1) represents a leaky process, where $u_o(t-\Delta t)$ is the membrane of neuron o at $t-\Delta t$ step, RC is a leaky constant. $U(t)$ is event signal from input neuron and other LIF neurons in the reservoir. Eq. (2) represents fire behaviour, in which \mathcal{G} is a step function, when $u_o(t) \geq V_{th}$, neuron o fires and generates a spike $s_o(t)$. V_{th} is the neuronal firing threshold voltage. Eq. (3) represents reset behaviour, if neuron o fires, the membrane resets to zero, otherwise it stays unchanged. In our experiments, the V_{th} is 0.18 V, and the RC is 20 μs .

The readout layer comprises a fully connected network of 27 neurons linked to eight output labels. These labels correspond to four different object shapes, each represented under high and low input intensities. This layer undergoes training using a backpropagation (BP) algorithm, adapted with a sigmoid-type surrogate gradient

function to accommodate the non-differentiable nature of spiking neurons. The learning process optimizes classification accuracy based on a cross-entropy loss function calculated from the spiking rates and the assigned labels of the output neurons. This structured approach ensures precise recognition of multimodal sensory data, substantiating the network's capability in handling complex sensory integrations.

Comment #2:

The SNOM results illustrate some change in signal in the channel area. However, SNOM signal is complicated and is dependent on temperature as well as the local optical properties of the film. What does the illustrated signal change in the channel area represent? Is it simply localized heating? Does it imply a change in phase? Do you have standardized signal of the VO₂ film at different temperatures that can be used to interpret the signal between the electrodes?

Response: Thank you for your insightful comments regarding the SNOM signal changes in the channel area. We acknowledge the complexity of SNOM signals in the VO₂ channel area. The SNOM technique, due to its sensitivity to both temperature and local optical properties, indeed presents a multifaceted response that could be influenced by various factors.

For signal interpretation in the channel area, the signal change illustrated in the SNOM images primarily represents changes in the local optical properties associated with the Insulator-Metal Transition (IMT) of the VO₂ channel area. We perform this SNOM experiment at mid-infrared (MIR) frequencies. *In-situ* SNOM mapping measurement of the device subjected to different voltage signals demonstrate the variation patterns of the SNOM signals in the channel area under the influence of distinct temperatures corresponding to different VO₂ channel currents. As shown in Figure R2, when the applied voltage (state ii) is smaller than the V_{th} (~4.0 V), the output current increases, and the device remains in a high resistance state (HRS). Although there is a variation in temperature within the channel area, it does not lead to significant changes in the SNOM signal of the VO₂ channel. Once the applied bias exceeds the V_{th} ,

the output current increases abruptly and the device transitions to a low resistance state (LRS) due to the phase transition from insulator to metal, which is accompanied by a change in the SNOM signal. This signal change is primarily due to the phase transition of the VO₂ film. Although there is a concomitant rise in channel temperature, the signal observed in SNOM image is predominantly attributable to modifications in the optical properties of the VO₂ film, resulting from the phase transition.

Figure R2. *In-situ* MIR-SNOM mapping measurement in VO₂ memristor. **(a)** *I-V* characteristics of VO₂ memristor with a 10 kΩ series resistor. **(b)** *In-situ* 3D optical amplitude map of the device area at different current values for the same device. The states of i and ii are in OFF state, iii is ON state.

Regarding the standardization of SNOM signals, the current SNOM measurement system we employed has limitations that prevent us from providing a systematic standardization for the SNOM signals of VO₂ films at different temperatures. However, we reference some reported work that has systematically studied the observation of VO₂ film phase transitions at different temperatures using SNOM^{R1-R3}. Stinson *et al.*^{R2} provided standardized SNOM signals of VO₂ films at different temperatures, as shown in Figure R3. The SNOM images reveal the near-field responses of VO₂ films at various temperatures, indicating that the change of the SNOM near-field signal between 340 and 350 K corresponds to a phase transition from an insulating to a fully metallic state. At around 341 K, the VO₂ sample phase-separates into metallic domains within an insulating background. By further analyzing the SNOM images, a significant jump in the MIR near-field signal is observed only when the temperature is below and above

the phase transition temperature. These results indicate that the SNOM signal primarily arises from changes in the optical properties of the VO₂ film induced by the phase transition.

Figure R3. SNOM images of the VO₂ IMT^{R2}. The images shown are taken at THz (top row) and MIR (bottom row) frequencies during a heating cycle. The temperature of each image is noted in the bottom left corner.

Figure R4. Analysis of the VO₂ SNOM images at THz and MIR frequencies^{R2}. The signal level S is shown normalized to that obtained on gold (S_M).

Corresponding references:

- R1. Qazilbash, M. M. et al. Mott transition in VO₂ revealed by infrared spectroscopy and nano-imaging. *Science* **318**, 1750-1753 (2007).
- R2. Stinson, H. T. et al. Imaging the nanoscale phase separation in vanadium dioxide thin films at terahertz frequencies. *Nat. Commun.* **9**, 3604 (2018).
- R3. Liu, M. K. et al. Anisotropic electronic state via spontaneous phase separation in strained vanadium dioxide films. *Phys. Rev. Lett.* **111**, 096602 (2013).

Following the reviewer's comments, we have added the discussion on SNOM signal

changes to the revised manuscript.

Revised main text:

Page 6 (Line 144-146): “Notably, the changes in the SNOM signal observed in the channel area are primarily associated with the phase transition of the VO₂ thin film. This transition involves significant alterations in the local optical properties of the material³⁴.”

Corresponding reference R2 has been added in the revised main text.

Revised supplementary information:

Page 9 (Line 96-102): “The states of i and ii correspond to the high resistance state (HRS) and low resistance state (LRS), respectively. When the applied voltage (state ii) is smaller than the V_{th} , the device remains in the HRS with minimal SNOM signal variation despite temperature changes in the channel area. Once the applied voltage exceeds the V_{th} , the device transitions to the LRS, and significant SNOM signal changes are observed. These changes are primarily driven by the phase transition in the VO₂ thin film, which alters its optical properties.”

Comment #3:

The results of stability with cycling (up to 10¹² cycles) are fairly impressive. Please clarify – were these cycles completed by sweeping voltage (with an external compliance current of 1 mA)? Or were these cycles completed within an LIF circuit? If the former, are these results fully translatable to endurance in an LIF circuit?

Response: Thank you for your interest in our results on the stability of the devices with cycling up to 10¹² cycles. We are pleased to provide clarification regarding the cycling process and its relevance to the endurance in an LIF circuit. The cycles mentioned were completed by applying voltage pulse across the VO₂ memristor under test. This method is a standard approach for evaluating the endurance and stability of devices. An external compliance current of ~2 mA was set to safeguard the memristor from overcurrent damage by incorporating a 1 kΩ resistor in series with the device.

To ensure the translatability of our results to an LIF circuit, we conducted further tests

within an actual LIF circuit configuration, as shown in Figure R5(a). The input voltage (V_{in}) is a constant voltage of 5 V. The test would involve subjecting the device to the operational parameters of the LIF circuit without capacitance to rapidly evaluate its endurance and stability. The number of switching was calculated by the following equation: $N = f_{firi} \times T$, where f_{firi} is the average oscillation frequency, which is greater than 500 kHz, and T is total oscillation time of 2.25×10^6 seconds, resulting in 1.1×10^{12} cycles. As shown in Figure R5(b), the memristor still work well even after 26 days of continuous oscillation. The oscillation cycle-to-cycle variation C_V in V_{th} and V_{hold} is as low as 2.26% and 3.05%, respectively, indicating the high robustness and stability of the device. Figure R4(c) show a stable oscillation behavior during 10^{12} cycles endurance test.

Figure R5. Endurance circuit measurement of the VO_2 memristor. **(a)** Illustration of proposed endurance test method with a memristor-based oscillator circuit and an oscilloscope (counter unit). V_{in} is set as 5 V, and R_L is set as 5 k Ω . **(b)** V_{th} and V_{hold} extracted from the oscillating behavior of the circuit. The memristor still work well

even after 26 days of continuous oscillation. The standard deviations of V_{th} and V_{hold} are 0.0432 V and 0.0106V, respectively. The coefficient of variation (C_v) in V_{th} and V_{hold} are as low as 2.26% and 3.05%, respectively. **(c)** Stable oscillating behavior of the VO₂ memristor circuit during 10^{12} switching cycles.

To address this point, we have revised the manuscript to include a more detailed description of the voltage sweeping process, and the considerations for translating these results to an LIF circuit environment.

Revised main text:

Page 6 (Line 152-157): “Notably, an external compliance current of ~2 mA was set to safeguard the memristor from overcurrent damage under applying voltage pulse by incorporating a 1 k Ω resistor in series with the device. The switching endurance of VO₂ memristor is translatable to stable oscillating behavior in a neuronal circuit, which further ensures the reliability of artificial neurons that incorporate such devices, detail results are shown in Supplementary Fig. 9.”

Revised supplementary information:

Figure R5 has been included in the supplementary information as **Supplementary Fig. 9**.

Page 10 (Line 113-115): “The number of switching was calculated by the following equation: $N = f_{fri} \times T$, where f_{fri} , the average oscillation frequency, is greater than 500 kHz, and T is total oscillation time of 2.25×10^6 seconds, resulting in 1.1×10^{12} cycles.”

Minor Points:

Comment #4:

Please review the manuscript completely for syntax grammar and spelling. E.g., line 59 “... when a human touch an object, ...”; line 79 “in-senor spike...”; line 82, “Third, it is required the flexibility in-sensor...”; etc. throughout.

Response: Thank you for your careful review and for identifying the spelling and grammatical errors in our manuscript. We have proofread the entire manuscript again

to ensure that any similar errors have been corrected. We have made the necessary corrections as outlined below:

Original Text	Corrected Text
(Page 3, line 59) “..., when a human touch an object, ...”	(Page 3, line 59) “..., when a person touches an object, ...”
(Page 3, line 79) “it is difficult to achieve crossmodal in-senor spike encoding”	(Page 3, line 79) “it is difficult to achieve crossmodal in-sensor spike encoding...”
(Page 3, line 81-84) “it is required the flexible in-sensor computing system to accurately identify multimodal objects and provide real-time haptic-feedback for the real application of in human-machine interaction.”	(Page 3, line 81-84) “ it is required that the flexible in-sensor computing system accurately identify multimodal objects and provide real-time haptic feedback for real applications in human-machine interaction. ”
(Page 4, line 87) “The CSSNs perform in-senor”	(Page 4, line 87) “The CSSNs perform in-sensor... ”
(Page 11, line 309) “our flexible system is capable of synchronously in-senor”	(Page 13, line 368) “our flexible system is capable of synchronously in-sensor ”
(Page 15, line 433-434) “A simple artificial LIF neurons are used to construct the virtual neurons, where the parameters were extracted from experimental data of VO ₂ -based LIF neurons.”	(Page 19, line 524-526) Simple artificial LIF neurons are used to construct the virtual neurons, where their parameters were extracted from experimental data of VO ₂ -based LIF neurons.
(Page 24, line 687) “ $R_1/R_{out} = 3 \text{ k}\Omega/50 \text{ }\Omega$ were used in the circuit”	(Page 28, line 808) “ $R_c/R_{out} = 3 \text{ k}\Omega/50 \text{ }\Omega$ were used in the circuit”

Comment #5:

Please clarify the orientation of the bending relative to the devices.

Response: Thank you for asking for more clarification on this. We have added a schematic diagram illustrating the orientation of the bending relative to the devices. As shown in Figure R6, this bending orientation is an out-of-plane bending (red arrows), meaning the device is flexing along a curvature.

Figure R6. A schematic diagram of the flexible device undergoing bending. The bending orientation is an out-of-plane bending (red arrows), meaning the device is flexing along a curvature.

To address this question, we have added the following sentences in the revised main text. The corresponding Figure R6 is also appended to Supplementary Fig. 11c in the supplementary information.

Revised main text:

Page 6 (Line 166-168): “**The bending orientation is an out-of-plane bending, meaning the device is flexing along a curvature (Supplementary Fig. 11c).**”

Comment #6:

Utilization of a 3D plot in figure 2f is not helpful, as it makes it difficult to discriminate the V_{th} at the different bending radii. I would recommend plotting this in a standard 2D plots with so the results at different bending radii can be directly compared.

Response: Thank you for your comment on the presentation of data in Fig. 2f. Following the suggestion, in this revised manuscript, we have replaced the 3D plot with a set of clear and concise 2D plot. This change facilitates a direct and easy comparison of the V_{th} values across different bending radii, as shown in below figure.

Revised main text (Page 27, Fig. 2f):

Comment #7:

The symbol illustrated in the LIF circuits (as well as the input in the video) implies that the externally applied voltage is applied in a pulse. Please clarify pulse length and duty factor of the input bias. What was the rationale for applying V_{in} in discrete pulses rather than in the constant steady state?

Response: Thanks for your inquiry regarding the pulsed voltage application in our LIF circuit experiments. The use of pulsed inputs in LIF circuit experiments is a widely adopted approach in many studies^{R4-R6}. In our experiment, we specifically used pulse lengths of 500/800 μ s with a duty factor of 33.3%. These parameters were carefully selected to match the temporal dynamics of the VO₂ neuron firing behavior. The choice of pulsed voltage application was based on several factors:

Temporal matching: The pulse length was chosen to align with the typical timescale of neural oscillatory activity. This alignment allows for a more realistic simulation of neuronal dynamics, ensuring that the temporal dynamics of the input match the temporal characteristics of neuronal firing behavior. The use of a longer pulse length also facilitates the calculation of the relationship between different inputs and neuronal firing frequencies in the experiment. Additionally, it allows the input voltage to be set to a constant value for an extended period, providing a stable input condition for analysis.

Energy efficiency: Pulsed inputs are more energy-efficient compared to constant

steady-state inputs. By applying voltage in discrete pulses, the system experiences periods of inactivity where no power is consumed, significantly reducing overall energy consumption. This energy efficiency is crucial in the design of neuro-inspired systems, particularly for applications requiring low power consumption.

Corresponding references

- R4. Wu, Q. et al. Spike encoding with optic sensory neurons enable a pulse coupled neural network for ultraviolet image segmentation. *Nano Lett.* **20**, 8015-8023 (2020).
- R5. Li, F. et al. A skin-inspired artificial mechanoreceptor for tactile enhancement and integration. *ACS Nano* **15**, 16422-16431 (2021).
- R6. Yuan, R. et al. A calibratable sensory neuron based on epitaxial VO₂ for spike-based neuromorphic multisensory system. *Nat. Commun.* **13**, 3973 (2022).

To enhance the transparency of the methods used, we have also added relevant applied pulse parameter in the main text and supplementary information.

Revised main text:

Page 28 (Line 804-809): “**a**, Circuit diagram of the artificial spiking temperature-sensing neuron, $V_{in} = 5 \text{ V}/800 \mu\text{s}$, $C_m = 10 \text{ nF}$, $R_L = 6.5/9.5 \text{ k}\Omega$, and $R_{out} = 50 \Omega$.” ...**d**, Circuit diagram of the artificial spiking pressure-sensing neuron. $V_{in} = 5 \text{ V}/800 \mu\text{s}$, $C_m = 10 \text{ nF}$, and $R_c/R_{out} = 3 \text{ k}\Omega/50 \Omega$ were used in the circuit.”

Revised supplementary information:

Page 17 (Line 178): “**(b)** The artificial neuron response under an input voltage (V_{in}) of $5 \text{ V}/800 \mu\text{s}$, $R_L = 5 \text{ k}\Omega$, $R_1 = 50 \Omega$, and $C_m = 10 \text{ nF}$.”

Page 35 (Line 365): “Bimodal pressure and temperature signals were simultaneously encoded to spikes by the CSSN under the conditions of 5.0 V input pulse amplitude, 500 μs interval, and 500 μs width.”

Comment #8:

Supplementary figure 4 suggests a potential oxygen gradient in the VO₂ layer. Can you

comment on this result and whether this reflects the existence of different oxidation states or is within instrumental resolution for the presented EDS results?

Response: Thank you for your insightful comment regarding a potential oxygen gradient in the VO₂ layer. We appreciate the opportunity to clarify our findings. Following your comment, we have re-examined our results and methodologies. Although the observed gradient might suggest varying oxidation states within the VO₂ layer, this interpretation is influenced by factors related to sample preparation and analysis techniques.

The main purpose of using EDS was used to qualitatively demonstrate the elemental distribution in our device structure. While powerful, EDS has limitations in spatial resolution and sensitivity to light elements like oxygen. The accuracy in quantifying oxygen can be affected by detector efficiency and FIB sample preparation artifacts. Moreover, in our initial EDS analysis, the TEM engineer overlooked the overlap between the L_α line (0.511 keV) of vanadium and the K_α line (0.53 keV) of oxygen^{R7},^{R8}, which may create an illusion of a gradient.

To address these concerns, we re-prepared the samples and conducted additional EDS testing, shown in Figure R7. The new EDS results demonstrate a homogeneous oxidation state within our VO₂ thin films.

Figure R7. Characterization of the VO₂ memristor. **(a)** Cross-sectional energy dispersive X-ray spectroscopy (EDS) mapping of Cr, V, and O elements in the device. **(b)** EDS elemental line profile of the device.

To further demonstrate the homogeneous valence distribution of the film, we performed valence distribution XPS profiling on a 100 nm-VO₂ thin film, etching the sample at various depths (20-80 nm). As can be seen from the Figure R8, there is no significant change in the valence distribution of the film as the etching depth increases. Figure R8(c) shows the distribution of V and O elements in the VO₂ film, and the atomic percentage of V and O does not change significantly with the increase of the etching depth. The XPS results further confirm no significant oxygen concentration gradient in the VO₂ layer.

Figure R8. The X-ray photoelectron spectroscopy (XPS) patterns of VO₂ films (100 nm) at different etch depths (20-80 nm).

Corresponding references:

- R7. Bearden, J. A. & Burr, A. F. Reevaluation of X-ray atomic energy levels. *Reviews of Modern Physics* **39**, 125-142 (1967).
- R8. Sivakumar, V., Suresh, R., Giribabu, K. & Narayanan, V. Solventless synthesis of m-LaVO₄ photocatalyst for the degradation of methylene blue and textile effluent. *Journal of Materials Science: Materials in Electronics* **28**, 4014-4019 (2016).

To address this question, we have revised the previous result, and added the new EDS experimental results in Supplementary Fig. 5.

Comment #9:

It would be helpful to clearly illustrate the impact of temperature on the response of the overall LIF circuit in fig. 3b (similar to that shown in fig. 3e for the effect of pressure).

Specifically, it would be helpful at this point to clearly illustrate that temperature impacts both frequency and amplitude of the resulting spikes.

Response: We appreciate the reviewer’s suggestion to illustrate the impact of temperature on the response of the overall LIF circuit.

To address this, we have revised Fig. 3b that clearly demonstrates the influence of temperature on the LIF circuit’s response.

Revised main text:

Page 8 (Line 219-226): “In this circuit, the operation temperature ranges from 21 to 42 °C is chosen to achieve effectively sensing-encoding under different load resistor (R_L). Real-time spiking responses are shown in Fig. 3b and Supplementary Fig. 18, 19. The firing frequency (f_{fi}) characteristics of this circuit under different temperatures and load resistor (R_L) are shown in Fig. 3c. As shown in Fig. 3b, when the external temperature rises from 21 to 42 °C, spike trains show an increased firing frequency (f_{fi}) and a reduced voltage amplitude.”

Fig. 3 | Temperature and pressure sensing and encoding characteristics of the artificial spiking sensory neuron. a, Circuit diagram of the artificial spiking temperature-sensing neuron, $V_{in} = 5$ V/800 μs , $C_m = 10$ nF, $R_L = 6.5/9.5$ k Ω , and $R_{out} = 50$ Ω . **b,** Real-time spiking responses from the spiking temperature sensory neuron by applying different temperature (21-42 °C). **c,** The f_{fi} of the spiking temperature sensory

neuron under different R_L and temperatures (21-42 °C). **d**, Circuit diagram of the artificial spiking pressure-sensing neuron. $V_{in} = 5 \text{ V}/800 \mu\text{s}$, $C_m = 10 \text{ nF}$, and $R_c/R_{out} = 3 \text{ k}\Omega/50 \Omega$ were used in the circuit. **e**, Real-time spiking responses from the spiking pressure sensory neuron by applying different pressures (0-18 kPa). **f**, Output firing frequency (f_{fri}) of the spiking pressure sensory neuron under different pressures. f_{fri} was obtained by averaging spike responses during six consecutive pressing-releasing cycles. The f_{fri} was obtained by averaging six spike responses at the same temperature.

Revised supplementary information:

Fig. 3b in the main text has been included in the supplementary information as **Supplementary Fig. 17a**.

Supplementary Figure 17. Temperature-dependent I - V switching characteristics of the VO_2 memristor under heating (a, b) and cooling (c). With increasing temperature, the window of $V_{th} - V_{hold}$ shrinks gradually. With decreasing temperature, the window of $V_{th} - V_{hold}$ expands gradually.

Comment #10:

On Figure 3d, clearly illustrate the piezo-resistor element (R_p) in the circuit.

Response: Thank you for your comment on enhancing the clarity of our figure. To clearly illustrate the components of the circuit, we have revised Fig. 3d.

Revised main text (Page 28, Fig. 3d):

Comment #11:

On suppl. Fig. 6 – you would not anticipate observing any peaks for VO_2 (R) (e.g., Shvets et al ‘A review of Raman spectroscopy of vanadium oxides’ 2019). However, small peaks are observed in figure b and c at ~ 160 and ~ 300 °C. Please comment on the origin and significance of these peaks. Does this imply phase impurity?

Response: Thank you very much for helping us find an area in the manuscript that was not rigorous enough. To address this question, we have re-extracted and enlarged the Raman spectra results at the highest temperatures (R-phase) from *Supplementary Fig. 6* and plotted them in the following figure.

Figure R9. Raman spectrum of VO_2 film at 81°C (Original Supplementary Fig. 6.).

A distinct peak at around 167 cm^{-1} can be observed, along with normal noise and a stronger signal peak from the Si substrate. After analysis, we identified two main reasons for this: (1) the samples used were exposed to air for nearly one month, leading

to surface oxidation; and (2) the surface of the sample was oxidized by the strong Raman laser during the test. These factors resulted in the appearance of a mixed phase of VO₂ and V₆O₁₃ on the surface of the film sample^{R9, R10}. To mitigate these issues, we used newly prepared samples and reduced the Raman laser power. The revised results, shown below (Figure R10), clearly demonstrate the M1-R phase transition, highlighting the excellent temperature sensitivity.

Figure R10. *In-situ* Raman spectrum of the VO₂ film during heating (a) and cooling (b) processes.

Corresponding references:

- R9. Shvets, P. et al. A review of Raman spectroscopy of vanadium oxides. *J Raman Spectrosc.* **50**, 1226-1244 (2019).
- R10. Zhang, C. et al. Characterization of vanadium oxide thin films with different stoichiometry using Raman spectroscopy. *Thin Solid Films.* **620**, 64-69 (2016).

Revised supplementary information:

To address this question, we have revised the previous Raman result into Supplementary Fig. 7.

Comment #12:

Similarly, in Suppl. Fig. 2 – there are minor XRD peaks at ~38, ~45 and ~57 deg 2theta. What are the origin of these peaks? Please index, and if possible, include a calculated

powder diffraction pattern or PDF pattern for reference.

Response: Thank you very much for your detailed comments. Following your suggestion, we have replotted the XRD peaks at ~ 37.5 , ~ 45 and ~ 57 in the Figure R11 and they matched well with the corresponding PDF card of monoclinic VO₂ (JCPDS No.43–1051).

Figure R11. X-ray diffraction (XRD) spectra of the VO₂ thin film grown on the Cr₂O₃ buffer layers.

2-Theta	d(Å)	I(f)	(h k l)
PDF#43-1051: VO ₂ Monoclinic P21/c			
27.795	3.2070	100.0	(0 1 1)
37.088	2.4220	30.0	(2 0 0)
39.714	2.2677	10.0	(0 2 0)
44.645	2.0280	6.0	(0 2 1)
57.424	1.6034	16.0	(0 2 2)
PDF#38-1479: Cr ₂ O ₃ Hexagonal, R-3c			
33.596	2.6653	100.0	(1 0 4)

Table R1. X-ray diffraction (XRD) peaks of VO₂ films grown on a Cr₂O₃ buffer layer compared to standard PDF cards.

Revised supplementary information:

To address this question, we have revised the previous result into **Supplementary Fig. 2b**, added the new table (**Supplementary Table 1**) into the revised supplementary information.

Page 33 (Line 299-304): “According to the XRD spectrum, the as-grown VO₂ film with buffer layer is polycrystalline, including the (011), (200), (020), (021), and (022) planes

of VO₂ (M), which has a structure with the P2₁/c space group (JCPDS card 43-1051), and the peak of (011) planes is significantly improved with the increase of buffer layer thickness, as shown in Supplementary Fig. 2b and Supplementary Table 1.”

Comment #13:

In Suppl. Fig. 5a, it is unclear what is the origin of the noise in the resistance signal at high temperatures (80 to 100 °C). This appears to be more of a result of some electrical noise or issues with the device, rather than an attribute of the phase transition process itself (this is a very peculiar signal which is not commonly observed in VO₂ devices).

Response: Thank you for your question; this is a very worthwhile discussion. The noise in the R-T curve is indeed, as you mentioned, an uncommon phenomenon for VO₂ thin films. With this data, we aimed to show that the electrical properties of the deposited VO₂ films are poorer when no buffer layer is present. To ensure the accuracy and reproducibility of our results, we prepared additional samples and conducted further experiments. The results consistently showed that the noise signal at high temperatures persisted, confirming that this noise is a reproducible phenomenon rather than an occasional occurrence, see Figure R12.

Figure R12. Temperature-dependent resistance of the VO₂ films without Cr₂O₃ buffer.

In fact, this noise problem is present in many poor-quality VO₂ films^{R11}. We believe that there are two main reasons for this:

First, the poor quality of the films grown by direct deposition at 280°C without a buffer layer results in a mixture of VO₂(M) and VO₂(B) phases. This mixed phase leads to

inhomogeneous phase transition behavior, which manifests as noise in the resistance signal.

Second, the low crystalline quality and roughness of the film, combined with poor contact with the test electrodes during surface resistance testing (especially at high temperatures), contribute to electrical noise. The lack of a buffer layer exacerbates the roughness issue, further increasing the noise.

Similar data are present in many works (Refs. R11-R13). It is worth mentioning that our buffer layer growth route effectively solves this problem, resulting in significantly reduced noise and improved electrical properties of the VO₂ films.

Corresponding references:

- R11. Cui, J. et al. Regulating the phase transition temperature of VO₂ films via the combination of doping and strain methods. *AIP Adv.* **13**, 055316 (2023).
- R12. Bhardwaj, D. et al. Synthesis of phase pure vanadium dioxide (VO₂) thin film by reactive pulsed laser deposition. *J. Appl. Phys.* **124**, 135301 (2018).
- R13. Polozov, V. et al. Thermally tunable frequency-selective surface based on VO₂ thin film. *Phys. Status Solidi A.* **217**, 2000452 (2020).

To address this question, we have added the following discussion into the Supplementary Fig. 6.

Revised supplementary information:

Page 7 (Line 68-74): “The noise in the resistance signal at high temperatures (80 to 100 °C) is likely due to electrical noise or device issues, rather than the phase transition process. This signal is not typical in high-quality VO₂ devices. For VO₂ films deposited without a buffer layer, the increased noise might be due to the mixture of VO₂(M) and VO₂(B) phases from direct deposition at 280°C, leading to inhomogeneous phase transition behavior, and low crystalline quality, increased surface roughness, and poor electrode contact at high temperatures may also contribute to this noise signal.”

Comment #14:

Specifics of the growth of the active VO₂ layer are not immediately clear. Under methods section in the main text, please expand upon (3) Cr₂O₃/VO₂ bilayer growth to at least include the type of target (V? VO₂?) and the partial pressure of the chamber during deposition.

Response: Thank you for your thorough review and constructive comments on our manuscript. We appreciate the opportunity to provide additional details regarding the growth of the Cr₂O₃/VO₂ bilayer.

In our study, the VO₂ films were deposited using a conductive V₂O₃ target (4 inches, 99.9% purity) with a 200W DC power supply. The air pressure in the sputtering chamber was maintained at approximately 8 mTorr using 49.15 sccm Ar (99.99% purity) and 0.85 sccm O₂ (99.99% purity) as the reaction gas. The Cr₂O₃ layer was deposited using a Cr₂O₃ ceramic target (4 inches, 99.9% purity) excited by a 150W RF power supply, with 40 sccm Ar (99.99% purity) as the reaction gas, ensuring a stoichiometric ratio. The air pressure in the sputtering chamber during the Cr₂O₃ deposition was about 6 mTorr, and both layers were deposited at 280 °C.

To further elaborate on the growth conditions of the Cr₂O₃/VO₂ bilayer, we have modified the descriptions in the device fabrication section of the main text.

Revised main text:

Page 17 (Line 478-483): “(3) Cr₂O₃/VO₂ bilayer films were grown on the flexible substrates by magnetron sputtering (Denton Discovery 635) at 280 °C. VO₂ films were deposited using a V₂O₃ target (4 inches, 99.9% purity) with a 200W DC power supply, maintaining pressure of 8 mTorr with 49.15 sccm Ar and 0.85 sccm O₂. The Cr₂O₃ layer was deposited using a Cr₂O₃ target (4 inches, 99.9% purity) with a 150W RF power supply and 40 sccm Ar under pressure of 6 mTorr.”

Reviewer #3 (Remarks to the Author):

The manuscript titled “Crossmodal sensory neurons based on high-performance flexible memristors for humans-machine in-sensor computing system” reported flexible VO₂ memristors to realize a crossmodal in-sensor computing system for wearable human-machine interfaces. The fabricated system enables real-time processing of multimodal signals, achieving high accuracy in object identification and real-time signal feedback. While the work is of interest to the field, results are somewhat of an improvement on what was previously published, so this reviewer thinks the novelty part is lacking for publication in this journal. Also, the manuscript must be carefully improved considering below points.

Response: Thank you very much for reviewing our manuscript. We have revised the manuscript to emphasize the novelty and which are also presented in responses to reviewer’s specific comments. Please kindly check. Below, we detail the unique aspects of our study, addressing your concerns about the originality and significance of our work.

1. **Record-High Performance in VO₂ Memristors:** For the first time, we report the achievement of record-high performance in the VO₂ memristor among current flexible neuronal switching devices. The flexible VO₂ (M) thin film, featuring high crystallization quality and a stable insulator-to-metal transition (IMT), is successfully deposited at a low temperature of 280°C by introducing a Cr₂O₃ buffer layer. This development has enabled us to fabricate a flexible forming-free volatile threshold switching (TS) memristor. The endurance of our device has been improved from 10⁸ to over 10¹² cycles, marking a significant enhancement of four orders of magnitude compared to previously reported state-of-the-art flexible TS devices (see Fig. 2g). Additionally, the VO₂ memristors show excellent performance in yield (~97.8%, 225 cells), high C2C and D2D uniformity (0.72% and 3.73%, respectively), speed (<30 ns), and flexibility (bendable to a radius of 1 mm). These achievements are unparalleled in existing flexible switching devices and facilitate reliable neuronal spike-encoding for constructing the CSSN.

2. **Innovative Flexible CSSN Hardware System:** We have designed and implemented a flexible CSSN hardware system for the first time. This system utilizes memristors to directly encode pressure and temperature signals into neuronal spiking signals, bypassing the need for complex analog-to-digital conversion modules. Real-time tactile feedback is achieved with this hardware system in human-computer interaction scenarios. In the flexible system, our CSSN operates within an energy range of 3.9-50 nJ per spike. This efficiency, combined with the ability to provide real-time tactile feedback, significantly enhances user experience and system energy efficiency in human-computer interaction scenarios.
3. **Crossmodal In-Sensor Spiking Reservoir Computing System:** We have achieved another pioneering breakthrough by constructing a crossmodal in-sensor spiking reservoir computing system, effectively addressing the training challenges of spiking neural networks, particularly with complex temporal information. Utilizing VO₂-based CSSN for in-sensor coding and VO₂-based LIF neurons for reservoir computing, our efficient multimodal processing neural network requires training only the readout layer. This system has demonstrated excellent results, achieving 97.8% accuracy in dynamic multimodal object recognition, and providing real-time haptic feedback. Compared to unimodal sensory recognition, our approach enhances recognition accuracy by approximately 25% and provides more realistic haptic feedback for harmful signals, significantly improving the utility and effectiveness of sensory recognition systems in practical applications.

To highlight the advances made in this work, we have made a thorough comparison between our work and other recent works, see Table R1 (Comment #6). This detailed comparative analysis thoroughly highlights the distinguishing features of various state-of-the-art in-sensor encoders, such as sensory signals, components, flexibility, endurance, energy efficiency, and specific applications.

Comment #1:

In the manuscript, the author claims that the device has forming-free characteristics.

However, there is no explanation as to why it has such characteristics. Additional explanation or analysis related to that must be provided.

Response: Thank you very much for the constructive comment.

The forming-free behavior in VO₂ devices can be attributed to the intrinsic metal-insulator transition (MIT) properties of VO₂ film. VO₂ undergoes a reversible phase transition from an insulating state to a metallic state at a relatively low temperature (around 68°C). However, achieving this requires relatively high-quality VO₂ films. In past studies, researchers have obtained phase transition characteristics in two primary ways:

High-Temperature Deposition: High-quality monoclinic (M)-phase VO₂ films require high-temperature annealing during formation (> 450 °C)^{R1-R3}. Devices based on M-phase VO₂ typically do not require a forming process.

Electroforming at Room Temperature: VO₂ films deposited at room temperature can form single-crystal VO₂ channels through an electroforming process^{R4, R5}. This process can create single-crystal VO₂ channels within the film without the need for additional high-temperature annealing steps.

Corresponding references:

- R1. Duan, Q. et al. Artificial multisensory neurons with fused haptic and temperature perception for multimodal in-sensor computing. *Adv. Intell. Syst.* **4**, 2200039 (2022).
- R2. Zhou, X., Gu, D., Li, Y., Qin, H., Jiang, Y. & Xu, J. A high performance electroformed single-crystallite VO₂ threshold switch. *Nanoscale* **11**, 22070-22078 (2019).
- R3. Yuan, R. et al. A calibratable sensory neuron based on epitaxial VO₂ for spike-based neuromorphic multisensory system. *Nat. Commun.* **13**, 3973 (2022).
- R4. Yeh, T.-H. et al. Enhancing threshold switching characteristics and stability of vanadium oxide-based selector with vanadium electrode. *IEEE Transactions on Electron Devices* **67**, 5059-5062 (2020).
- R5. Xue, W. et al. A 1D vanadium dioxide nanochannel constructed via electric-

field-induced ion transport and its superior metal–insulator transition. *Adv. Mater.* **29**, 1702162 (2017).

In our study, we aimed to achieve excellent MIT features while maintaining a lower deposition temperature suitable for flexible electronics manufacturing. To overcome the challenge of high-temperature requirements, Cr₂O₃ was chosen as a buffer layer to lower the lattice mismatch between VO₂ and SiO₂ and to reduce the deposition temperature of the VO₂ film to 280 °C. It was found that the Cr₂O₃ buffer layer helps to improve the crystalline quality of the VO₂ (M) thin film, resulting in a stable MIT process. Without the buffer layer, the crystallinity of the deposited VO₂ films is insufficient, necessitating a forming process to create a VO₂ phase transition channel, see Figure R1. This formed channel can be random and lead to device instability. The high crystalline quality of VO₂ also leads to markedly improved cycle-to-cycle (C2C) uniformity. With the increase in buffer layer thickness, the C2C variability in V_{th} decreases from 4.47% to 0.51%. The VO₂ memristor based on 40-nm Cr₂O₃ demonstrates extremely low C2C variability due to its high crystalline structure.

Figure R1. I - V characteristics of the VO₂ devices grown with Cr₂O₃ buffer layer of (a) 0 nm, (b) 10 nm, (c) 20 nm, (d) 30 nm, and (e) 40 nm. (f) Cycle-to-cycle V_{th} distribution of VO₂ device with different thickness of Cr₂O₃ buffer layer. For different devices, 50 repeated switching cycles were measured.

To address this point, we have added an explanation about the forming-free characteristic in the text and provided experiment results where a forming process was observed.

Revised main text:

Page 5 (Line 133-135): “The forming-free behavior is primarily attributed to the high-quality VO₂ (M) thin films obtained by introducing a Cr₂O₃ buffer layer with optimal thickness, as shown in Supplementary Fig. 3.”

Revised supplementary information:

The corresponding Figure R1 is also appended in the Supplementary Fig. 3 in the supplementary information.

Page 8 (Line 45-51): “Without the buffer layer, the crystallinity of the deposited VO₂ films is insufficient, necessitating a forming process to create a VO₂ phase transition channel. This formed channel can be random and lead to device instability. The high crystalline quality of VO₂ also leads to markedly improved C2C uniformity. With the increase in buffer layer thickness, the C2C variation in V_{th} decreases from 4.47% to 0.51%. The VO₂ memristor based on 40-nm Cr₂O₃ demonstrates extremely low C2C variability due to its high crystalline structure.”

Comment #2:

It is known that materials-based threshold switching devices with MIT (Metal-Insulator Transition) characteristics, such as NbO₂, exhibit significant switching behavior variations depending on the active area size (the distance between in planar device). What is the active area size of the manufactured device or the distance between the electrodes of the planar device? To further validate the stability and scalability of the device, additional experiments should be conducted to observe how its properties evolve with changes in the active area size or the distance between electrodes.

Response: Thank you very much for your insightful comments regarding the active area size of our device. Your suggestion to investigate the stability and scalability of the device with respect to the active area size is well-taken.

For the active area size of the VO₂ memristor, the device was designed as a planar structure with a channel length of 500 nm and a width of 1 μm, as shown in Fig. R2.

Figure R2. SEM image of the active area size in planar VO₂ memristors.

To further address the stability and scalability of the devices, we conducted additional experiments to assess the TS behavior across devices with varying channel lengths (300 nm to 4 μm) and widths (500 nm to 4 μm), as shown in Figure R3. Each device's TS behavior was tested consistently 10 times. The variation in device dimensions affects the total area over which the IMT must propagate, subsequently impacting the threshold switching characteristics.

Our results indicate that the pristine device resistance (R_{off}) increases with increasing channel lengths and decreases with increasing channel widths, as shown in Figure R3b. This trend indicates the channel area (width × length) significantly affects R_{off} . An increase in R_{off} is accompanied by a higher V_{th} , as more energy is required to achieve the same level of Joule heating necessary to induce the IMT. The relationship between V_{th} and R_{off} can be described by the following equation^{R3}:

$$V_{\text{th}}(T_0) = \sqrt{\frac{R_{\text{off}}}{R_{\text{th}}}}(T_t - T_0) \quad (\text{R1})$$

where R_{th} , T_t , and T_0 are the effective thermal resistance, the transition temperature of VO₂ and the operating temperature, respectively. Thus, the V_{th} and R_{off} are positively correlated.

These observed results provide clear evidence of the excellent scalability of the TS performance, supporting the robustness and reliability of these devices for potential applications.

Figure R3. Threshold switching behavior of the flexible planar VO₂ device under different active area size. **(a)** The I - V curves for different device size, with channel lengths from 300 nm to 4 μm and widths from 500 nm to 4 μm. Each size was tested for 10 cycles, exhibiting stable threshold switching behaviors. Pristine device resistance (R_{off}) **(b)** and V_{th} **(c)** for device with varying channel lengths and widths.

We revised the main text and supplementary information to include the results of these additional experiments.

Revised main text:

Page 4 (Line 114): “The active area size of the VO₂ memristor is a channel length of 500nm and a width of 1 μm.”

Page 7 (Line 191-198): “The scalability of the flexible device is critical for a wide range of applications, particularly in the fields of scalable electronics and large-area sensing systems. To investigate the scalability of the flexible VO₂ memristor, I - V characteristics of the devices with different sizes were measured across a range of channel lengths (300 nm to 4 μm) and widths (500 nm to 4 μm), as shown in Supplementary Fig. 15. The

results demonstrate that the V_{th} decreases as the device size decreases, and there is no significant change in the TS stability of the device, indicating that the device is scalable without property degradation.”

Revised supplementary information:

The corresponding Figure R3 has been appended to **Supplementary Fig. 15** in the supplementary information.

Page 16 (Line 166-172): “The change in device area affects the total area over which the IMT must propagate and affects the TS performance. The R_{off} increases with increasing channel lengths and decreases with increasing channel widths. V_{th} will increase along with higher R_{off} , as more energy is required to achieve the same level of joule heating necessary to induce the IMT. Consequently, a higher V_{th} is observed. These results demonstrate a design space and pathway toward realizing low-power operation and high-density integration.”

Comment #3:

What is the thickness of the buffer layer proposed in the manuscript, and is there any change in the crystallinity or device characteristics of the VO₂ thin film depending on the buffer layer thickness?

Response: We greatly appreciate the reviewer’s insightful comments. In our study, the buffer layer thickness is designed to be 40 nm, which has been determined to be optimal for providing high crystallinity to the VO₂ film at low growth temperatures.

Following the reviewer’s suggestions, we have fabricated VO₂ memristors with buffer layer thicknesses ranging from approximately 0 nm to 40 nm. According to the X-ray diffraction (XRD) spectrum, the as-grown VO₂ film with buffer layer is polycrystalline, including the (011) and (020) planes of VO₂ (M), which has a structure with the P2₁/c space group (JCPDS card 72-0514), and the peak of (011) planes is significantly improved with the increase of buffer layer thickness, see Figure R4 (a). The transmission electron microscopy (TEM) images are presented in Figure R4 (b, c). The crystallinity of the VO₂ films deposited on a 40 nm-Cr₂O₃ buffer layer is significantly

improved compared to the one without buffer layer. These results indicate the Cr_2O_3 buffer layer promotes the crystallization of VO_2 (M) at low growth temperature. This enhancement in VO_2 crystallinity is crucial as it forms the basis for the stable phase transition characteristics of the device.

Figure R4. Characterization of the Cr_2O_3 film and VO_2 film. (a) X-ray diffraction (XRD) spectra of the VO_2 thin film grown on the Cr_2O_3 buffer layers. Transmission electron microscopy (TEM) images and electron diffraction patterns of the VO_2 film without (b) /with (c) Cr_2O_3 layer.

Furthermore, devices with poorer crystallinity may undergo an electroforming process to establish a conductive path in VO_2 , whereas devices with high crystallinity can operate forming-free. This forming-free operation is important for device reliability and uniformity (see Figure R5). The high crystalline quality of VO_2 also leads to markedly improved cycle-to-cycle (C2C) uniformity. With the increase in buffer layer thickness, the C2C variability in V_{th} decreases from 4.47% to 0.51%. The VO_2 memristor based on 40-nm Cr_2O_3 demonstrates extremely low C2C variability due to its high crystalline structure.

Figure R5. I - V characteristics of the VO₂ devices grown with Cr₂O₃ buffer layer of (a) 0 nm, (b) 10 nm, (c) 20 nm, (d) 30 nm, and (e) 40 nm. (f) Cycle-to-cycle V_{th} distribution of VO₂ device with different thickness of Cr₂O₃ buffer layer. For different devices, 50 repeated switching cycles were measured.

These findings highlight the significant role of the Cr₂O₃ buffer layer in enhancing the crystallinity and device performance of VO₂ thin films. The optimal buffer layer thickness of 40 nm ensures high crystalline quality, stable phase transition characteristics, and low C2C variability, thereby contributing to the superior performance of our Cr₂O₃/VO₂-based memristor devices.

To address this question, we have added the new experimental results in supplementary information and added the following discussion into the revised manuscript.

Revised main text:

Page 5 (Line 122-129): “We optimized the buffer layer’s thickness by preparing VO₂ thin films and devices with varying buffer layer thicknesses (0-40 nm), achieving an optimal thickness of 40 nm, as shown in Supplementary Fig. 2-7. It is found that the Cr₂O₃ buffer layer helps to improve the crystalline quality of the VO₂ (M) thin film, resulting in a stable MIT process, detail results are explained in Supplementary Note 2. The VO₂ memristor with a 40 nm-thick buffer layer not only exhibits forming-free behavior but also achieves high stability and uniformity. These improvements promote

practical applications and large-scale fabrication.”

Revised supplementary information:

The corresponding Figure R4 and Figure R5 have been included in the supplementary information as **Supplementary Fig. 2, 3**, respectively.

Comment #4:

The author claimed that it mimics the human sensitization function by using the example of responding even when a small pressure is applied when exposed to high temperatures. However, Sensitization typically entails responding to the same non-harmful stimulus even upon sustained exposure. The author’s claim about sensitization imitation should be presented in more detail or examined more closely.

Response: Thank you very much for your insightful comment. We sincerely apologize for any confusion caused by our initial explanation on the concept of “human sensitization” and appreciate the opportunity to clarify and expand upon this important aspect of our research.

We acknowledge the reviewer’s concern that sensitization typically involves responding to the same non-harmful stimulus. In biology, sensitization refers to a physiological phenomenon where an organism’s response to a stimulus becomes more pronounced after repeated exposure. This phenomenon can be observed in various biological systems, including the nervous and immune systems. In our study, we aimed to emulate key features of sensitization observed in biological nociceptors. The sensitization process involves hyperalgesia, an intensified sensitivity to detrimental stimuli, as well as allodynia, where pain is caused by normally harmless stimuli^{R6, R7}. For instance, when skin is wounded and turns into a bruising state, it becomes highly responsive to ensuing stimuli, even a gentle touch, as a protective action to prevent further harm to the injured tissue. We investigated the sensitization properties of our artificial nociceptors by varying temperature and pressure. When our device is exposed to a higher temperature, it undergoes a state change similar to how biological neurons become more responsive after harmful stimulation.

In light of your feedback, we have provided more detailed results of the CSSN response under varying pressure and temperature, which simulate the “sensitization” properties of the artificial nociceptors. We applied different temperature to the VO₂ devices, introducing a change that emulates the injury to the artificial nociceptor system. As shown in Figure R6, the output spike frequency (f_{fri}) recorded under varying input pressures for devices that have experienced different level of “injury”. It is clear that the “injured” CSSN had a higher f_{fri} applying the same pressure, featuring the hyperalgesia characteristic. Notably, the pressure threshold shifted lower due to the V_{th} of the device decreasing as temperature increase (Figure R6(b)). Compared to the normal state, a smaller pressure threshold can activate the injured CSSN, reproducing the allodynia characteristics. These results illustrate that our CSSN successfully emulates the sensitization characteristics of the biological nociceptors.

However, we acknowledge that the precise emulation of biological sensitization mechanisms can be challenging due to the complex interplay of molecular and cellular changes that occur in biological systems, which may not be fully replicable with current electronic components.

Figure R6. Basic demonstration of the multimodal nociceptor neuron. **(a)** The sensitization in the artificial nociceptors characterized by hyperalgesia and allodynia. **(b)** Threshold voltage of the device at different injury state.

Corresponding references:

R6. Gold, M. S. & Gebhart, G. F. Nociceptor sensitization in pain pathogenesis. *Nat. Med.* **16**, 1248-1257 (2010).

R7. Jensen, T. S. & Finnerup, N. B. Allodynia and hyperalgesia in neuropathic pain: Clinical manifestations and mechanisms. *Lancet Neurol.* **13**, 924-935 (2014).

To address this point, we have added relevant discussions in the main text and **Supplementary Note 3** of the revised manuscript.

Revised main text:

Page 12 (Line 337-339): “This phenomenon is similar to the “sensitization” of biological nociceptor neurons under injury, as elaborated in Supplementary Fig. 24 and Supplementary Note 3.”

Revised supplementary information:

The corresponding Figure R6 has been appended as **Supplementary Fig. 24** in the supplementary information.

Supplementary Note 3: “Sensitization” of the artificial nociceptor under injury.

In biological nociceptor, sensitization is a physiological phenomenon that enhances the pain sensitivity of sensory neurons to noxious stimuli^{20,21}. This often manifests in two primary forms: allodynia, where a normally innocuous stimulus becomes painful, and hyperalgesia, where the response to a noxious stimulus becomes more pronounced. Allodynia is characterized by a decreased response threshold to the stimulus, while hyperalgesia is marked by an increased response intensity to the same stimulus. For example, when skin is wounded and becomes bruised, it becomes highly responsive to subsequent stimuli, even a gentle touch, as a protective action to prevent further harm to the injured tissue.

To demonstrate the “sensitization” feature of our CSSN, we applied different temperature to the VO₂ devices, simulating “injury” to the artificial nociceptor system. As shown in Supplementary Fig. 24a, the output f_{fi} under different input pressures was recorded for devices that experienced various levels of “injury”. The results show that the “injured” CSSN exhibited a higher f_{fi} for the same applied pressure, indicating the hyperalgesia characteristic. Notably, the pressure threshold shifted lower because the V_{th} of the device decreased with increasing temperature (Supplementary Fig. 24b). In comparison to the normal state, a smaller pressure threshold could activate the injured

CSSN, replicating the allodynia characteristic. Interestingly, the damage to the artificial nociceptor is recoverable because the VO₂ device's damage (cooling process, Supplementary Fig. 17c) is reversible, corresponding to a transient skin allergy.

Corresponding references R6, R7 have been added in the revised supplementary information.

Comment #5:

Figure 4c shows a tendency for the maximum amplitude voltage of a spike to decrease as the stimulus (pressure) becomes stronger. As pressure increases, the voltage generated by the pressure sensor is expected to increase; thus, the applied voltage will also increase. Why does the absolute amplitude tend to decrease?

Response: Thank you for your thoughtful comment. We acknowledge that this aspect was not clearly explained in our initial manuscript. To address your question, we have re-extracted and enlarged the CSSN's spike response results at 24 °C from Fig. 4c and plotted them in the following Figure R7.

Figure R7. Temperature and pressure sensing and encoding characteristics. (a) Real-time spiking responses of the CSSN under co-stimulation of pressure (7-18 kPa) and temperature (24-42 °C). (b) Spiking responses of CSSN to different pressure at 24 °C.

As shown in Figure R7(b), the spike amplitude voltage does not decrease with increasing pressure. To clarify this, we will explain the specific trends in our data in more detail. CSSN synchronously encode pressure and temperature signals into neuronal spiking signals with different spike frequency (f_{fir}) and amplitude (V_{out}). The spike amplitude is predominantly influenced by the V_{th} of VO₂-based memristors. As

the applied pressure and temperature increases, the resistance of the piezoresistive sensor and the V_{th} decreases. This causes an increase in f_{fri} and a decrease in V_{out} . According to the analysis of the neuronal circuit, the V_{out} is related to V_{th} , monitor resistor (R_{out}), and on-state resistance of the memristor (R_{on}) as follows:

$$V_{out} = V_{th} \times \frac{R_{out}}{R_{out} + R_{on}} \quad (R2)$$

Although an increase in pressure results in a lower resistance (R_p) generated by pressure sensor, the V_{th} does not change with different pressures, as shown in Figure R8(a).

Furthermore, the V_{th} of the VO₂-based memristor is sensitive to temperature changes in the crossmodal coding process. As shown in Figure R8(b), the V_{th} of VO₂-based memristor decreases with increasing temperature, regardless of the applied pressure. This temperature-dependent behavior of the V_{th} provides a regulatory mechanism for encoding thermal information into the spike amplitude. The coupling of various pressure signals through our CSSN, in conjunction with the temperature signals, results in the observed decreasing trend in V_{out} . This intricate interplay between pressure and temperature inputs highlights the system's sophisticated ability to encode and decouple multimodal information.

Figure R8. Variation curves of the V_{th} (a) and the output spike amplitude (V_{out}) (b) at different pressures and temperatures.

To address this question, we have included the results in supplementary information and added a brief discussion in the revised manuscript.

Revised main text:

Page 12 (Line 316-323): “This phenomenon occurs due to the CSSN’s dynamic

response to multimodal inputs. Specifically, an increase in pressure generates a lower resistance at the pressure sensor, which is transduced into a higher f_{fir} . The amplitude of the output spikes is predominantly influenced by the V_{th} of the VO₂-based memristor, which is inherently sensitive to thermal changes. As depicted in Supplementary Fig. 22a, as temperature rises, there is a corresponding decrease in the V_{th} of the memristor. According to Eq. 1, this decrease in V_{th} consequently results in a reduced V_{out} (Supplementary Fig. 22b).”

Revised supplementary information:

The corresponding Figure R8 has been appended to **Supplementary Fig. 22** in the supplementary information.

Comment #6:

The proposed CSSN-based robot arm is very interesting due to its demonstration of the potential of device applications. However, there appears to be a deficiency in elucidating the device's novelty. The author proposed a comparison with various devices in the supplementary Table 1, but additional review is needed. The paper below, which implements multimodal sensing based on the same material (VO₂-based memristor) proposed, should be reviewed, and the novelty of the device should be additionally explained.

*“Duan, Qingxi, et al. "Artificial multisensory neurons with fused haptic and temperature perception for multimodal in-sensor computing." *Advanced Intelligent Systems* 4.8 (2022): 2200039.”*

Response: Thank you for your valuable feedback and for recognizing the potential of our work. We appreciate the reviewer bringing Duan et al. (2022) to our attention. Their work is indeed excellent, with comprehensive analysis and results. However, our flexible sensory neurons differ significantly from those reported by Duan et al.

Following your suggestion, we have thoroughly reviewed Duan et al. (2022) and other relevant studies to provide a detailed comparison that underscores the unique contributions and technological advancements of our system.

Detailed Comparison with Duan et al. (2022):

1. Device Performance:

- **Duan et al. (2022):** Their research utilizes high-temperature epitaxial growth at 530°C to deposit high-crystalline quality VO₂ thin films, achieving superior memristor performance such as high endurance (>10⁶ cycles), fast response time (<120 ns), and excellent stability (C2C V_{th} distribution: 1.36–1.46 V across 100 repeated DC cycles, D2D V_{th} distribution: 1.3–1.56 V across 10 cells). However, the high-temperature requirement limits the use in flexible electronics and wearable devices.
- **Our Work:** We introduced a Cr₂O₃ buffer layer, enabling the deposition of high-performance VO₂ memristors at a significantly lower temperature of 280°C. This innovation maintains forming-free behavior and achieves excellent yield (97.3%, 225 cells), ultrahigh endurance (>10¹² cycles), low variability in V_{th} (C2C distribution: 1.73–1.75 V across 2000 repeated DC cycles; D2D distribution: 1.73–1.95 V across 220 cells), a reduced response time (<30 ns), and flexibility (bendable to a radius of 1 mm). The endurance of our device marks a significant enhancement of six orders of magnitude compared to the VO₂ devices reported by Duan et al. Our approach allows for the large-scale production of flexible memristors, expanding their potential in flexible and wearable electronics. This innovation of this low temperature method also benefits broader CMOS electronic applications.

2. Neuron Functionality:

- **Duan et al.:** This study integrates VO₂-based memristors with piezoresistive sensors for multimodal sensing, focusing on the fusion of pressure and temperature inputs for in-sensor encoding. They conducted a detailed study of the temperature and pressure encoding properties of multimodal neurons.
- **Our Work:** We implemented a hardware-integrated flexible multimodal sensory neuron. This flexible CSSN not only efficiently encodes but also decodes multimodal information, showing advanced data compression and

conversion capabilities. Additionally, our CSSN employs a crossmodal sensory and encoding strategy that mimics advanced biological proprioceptive reflexes, essential for damage prevention in sensory robotics and smart interfaces.

3. Application Scope:

- **Duan et al.:** This work primarily explores the potential of VO₂ neurons in multimodal pattern recognition. Using a multi-layer perceptron network, they achieve an accuracy of 91.35% in multimodal object recognition after 200 epochs.
- **Our Work:** We extend the application of flexible CSSN to robotic systems, integrating them into robotic arms that actively respond to sensory inputs. This facilitates real-time interaction and adaptive responses to environmental stimuli. Moreover, we are the first to construct a crossmodal in-sensor spiking reservoir network, requiring training only the readout layer, and achieve an impressive accuracy of 98.1% for dynamic multimodal object recognition and real-time sensory feedback. It is crucial for enhancing tactile intelligence and precise recognition in various applications, including flexible electronics and humanoid robotic systems.

Expanded Comparative Analysis:

In response to your comments, we have broadened our comparison to include both flexible and rigid substrate-based in-sensor spike encoding neuron, covering both single-modal and multi-modal configurations, as depicted in Table R1. This detailed comparative analysis highlights the distinguishing features of various state-of-the-art in-sensor encoders, including sensory signals, components, flexibility, endurance, energy efficiency, and specific applications.

Generally, most of the state-of-the-art in-sensor spike sensory neuron are single-modal in-sensor spike encoder inspired by one of the human senses, as demonstrated in the table below. Additionally, these encoders use memristors/transistors on rigid substrates, making it difficult to construct flexible integrated sensory-computing systems. These encoders demonstrate outstanding performance in terms of energy efficiency and

diverse applications. Potential applications predominantly include edge detection, image segmentation, pattern recognition, neuromorphic bio-interfaces, motion detection, and machine vision.

In contrast, our crossmodal in-sensor spike neuron exhibits several important features distinct from recently reported works. First, unlike typical single-modal systems that mimic a single human sense, our system integrates tactile and temperature stimuli, offering richer and more diverse sensory input akin to the human somatosensory system. Second, our flexible crossmodal encoder design adopts a compact and flexible 1M+1PS architecture, with the flexible memristor achievable through large-area and scalable fabrication techniques. This enables the creation of portable and wearable sensory computing systems. Third, our encoder achieves endurance exceeding 10^{12} cycles and operates within an energy range of 3.9-50 nJ per spike, highly competitive within the field. These features contribute to high stability and low power consumption, facilitating the development of energy-efficient sensory computing systems. Fourth, our encoder’s broad application potential ranges from human-machine interaction to dynamic object recognition, crucial for advancing tactile intelligence and damage prevention in sensory robotics and smart interfaces. We hope this pioneering effort will pave the way for future advancements in the field. Therefore, our work significantly differentiates from most existing works in this field.

Sensory signals	Crossmodal sensory	Sensory components	Flexibility	Endurance	Energy/spike	Coding-related applications	Ref.
Optical	No	1M	No	>100	2.1-20.3 nJ	Machine vision	R8
Optical	No	1M	No	> 10^4	~190 nJ*	Image segmentation	R9
Temperature	No	1M	No	> 10^3	~0.15 nJ*	Edge detection	R10
Physiological signals	No	2T	No	/	~0.5 μ J*	Neuromorphic bio-interface	R11
Optical	No	1M	No	>500	~32 pJ*	Pattern recognition	R12
Pressure & Temperature	Yes	1M+1PS	No	> 10^6	/	Object recognition	R1
Pressure & optical	Yes	1T+1PS	No	/	~8 nJ*	/	R13
Optical	No	1M+1T	No	> 10^8	~0.1 nJ	Motion detection	R14
Pressure & Temperature	Yes	1M+1PS	Yes	> 10^{12}	3.9-50 nJ	Human-machine interaction & dynamic object recognition	This work

*The energy consumption per spike is calculated approximately from the P-t, V-t and I-t curves in these reference papers, respectively. To unify the benchmark, all the sensory components in these reference papers are equivalent to three categories: memristor (M), transistor (T), and pressure sensor (PS).

Table R1. Comparison with the state-of-the-art in-sensor encoding neurons.

Corresponding references:

- R8. Wang, Y. et al. Memristor-based biomimetic compound eye for real-time collision detection. *Nat. Commun.* **12**, 5979 (2021).
- R9. Nath, S. K. et al. Optically tunable electrical oscillations in oxide-based memristors for neuromorphic computing. *Adv. Mater.* 2400904 (2024).
- R10. Shi, K. et al. An oxide based spiking thermoreceptor for low-power thermography edge detection. *IEEE Electron Device Lett.* **43**, 2196-2199 (2022).
- R11. Sarkar, T. et al. An organic artificial spiking neuron for in situ neuromorphic sensing and biointerfacing. *Nat. Electron.* **5**, 774-783 (2022).
- R12. Wang, X. et al. Vertically integrated spiking cone photoreceptor arrays for color perception. *Nat. Commun.* **14**, 3444 (2023).
- R13. Sadaf, M. U. K., Sakib, N. U., Pannone, A., Ravichandran, H. & Das, S. A bio-inspired visuotactile neuron for multisensory integration. *Nat. Commun.* **14**, 5729 (2023).
- R14. Wang, R. et al. 1-phototransistor-1-threshold switching optoelectronic neuron for in-sensor compression via spiking neuron network. In *2023 International Electron Devices Meeting (IEDM)* 1-4 (2023).

Following the reviewer's comments, we have created a table (**Table 1**) comparing the results of various in-sensor spike encoder. And, we added a section discussing the comparison associated with our flexible CSSN in the revised manuscript.

Revised main text:

Page 16 (Line 448-458): “We have provided a comprehensive overview of recent advancements in artificial in-sensor spike encoding neurons^{25,29,57-62}, summarized in Table 1. Our CSSNs offer several advantages, including flexibility that support multimodal sensory inputs like tactile and temperature stimuli, emulating the human somatosensory system. Moreover, CSSNs demonstrate excellent endurance exceeding 10^{12} cycles and operate within an energy range of 3.9-50 nJ per spike. This positions them favorably in terms of energy efficiency and durability compared to other state-of-

the-art in-sensor spike encoders. CSSNs support a wide range of applications, including dynamic object recognition and human-machine interactions. They are ideally suited for creating compact, versatile wearable sensory computing systems, potentially revolutionizing the landscape of portable and wearable technology.”

Corresponding references R1, R8-R14 have been added in the revised main text.

Reviewer #4 (Remarks to the Author):

Response: Thank you very much for your participation in the peer review process. We appreciate the collaborative effort in reviewing our manuscript and the valuable insights provided.

REVIEWERS' COMMENTS

Reviewer #1 (Remarks to the Author):

The revised manuscript has been significantly improved with additional results. The new version can be recommended for publication.

Reviewer #2 (Remarks to the Author):

The authors have sufficiently responded to all of my previous comments. I will reiterate my initial impression of this manuscript.

Overall, the manuscript is very well supported with sufficient supporting evidence and documentation. This manuscript demonstrates a significant advance both in terms of the repeatability and reliability of the low-T array of VO₂ devices on a flexible substrate, as well as its implementation in a sensory feedback scheme. Specifically, the high yield, robust behavior of LIF devices based on VO₂ (even despite depositing on a flexible substrate and subjecting to bending/stretching) represents a significant improvement over prior demonstrations in the area. Furthermore, the combined dual-use function is intriguing and may beget further novel device concepts. Based on the merits of these contributions, this is an important contribution and merits publication.

I believe that the modifications to the manuscript made in response to my previous comments (as well as other reviewers' comments) are appropriate and sufficient, and have significantly clarified a number of elements of this manuscript. Specifically, the methodology appears sound, and the study is reproducible.

Reviewer #3 (Remarks to the Author):

This reviewer has carefully read the comments from other reviewers and authors answers. This reviewer admits that the authors made considerable effort to answer and revise their manuscript following the reviewers' comments. In terms of the novelty, this reviewer thinks that the current work is not a very new one but still there is an significant advances compare to the previous works. Therefore, this reviewer recommends accepting this manuscript at this stage.

Reviewer #4 (Remarks to the Author):

"I co-reviewed this manuscript with one of the reviewers who provided the listed reports. This is part of the Nature Communications initiative to facilitate training in peer review and to provide appropriate recognition for Early Career Researchers who co-review manuscripts."

Manuscript ID: NCOMMS-24-13684A

Reviewer #1 (Remarks to the Author):

The revised manuscript has been significantly improved with additional results. The new version can be recommended for publication.

Response: Thank you very much for recommending the publication of our manuscript. We sincerely appreciate your valuable time and effort in reviewing our work.

Reviewer #2 (Remarks to the Author):

The authors have sufficiently responded to all of my previous comments. I will reiterate my initial impression of this manuscript.

Overall, the manuscript is very well supported with sufficient supporting evidence and documentation. This manuscript demonstrates a significant advance both in terms of the repeatability and reliability of the low-T array of VO₂ devices on a flexible substrate, as well as its implementation in a sensory feedback scheme. Specifically, the high yield, robust behavior of LIF devices based on VO₂ (even despite depositing on a flexible substrate and subjecting to bending/stretching) represents a significant improvement over prior demonstrations in the area. Furthermore, the combined dual-use function is intriguing and may beget further novel device concepts. Based on the merits of these contributions, this is an important contribution and merits publication.

I believe that the modifications to the manuscript made in response to my previous comments (as well as other reviewers' comments) are appropriate and sufficient, and have significantly clarified a number of elements of this manuscript. Specifically, the methodology appears sound, and the study is reproducible.

Response: Thank you very much for recommending the publication of our manuscript. We sincerely appreciate your valuable time and effort in reviewing our work.

Reviewer #3 (Remarks to the Author):

This reviewer has carefully read the comments from other reviewers and authors answers. This reviewer admits that the authors made considerable effort to answer and revise their manuscript following the reviewers' comments. In terms of the novelty, this reviewer thinks that the current work is not a very new one but still there is an significant advances compare to the previous works. Therefore, this reviewer recommends accepting this manuscript at this stage.

Response: Thank you very much for recommending the publication of our manuscript. We sincerely appreciate your valuable time and effort in reviewing our work.

Reviewer #4 (Remarks to the Author):

Response: Thank you very much for your participation in the peer review process. We appreciate the collaborative effort in reviewing our manuscript and the valuable insights provided.